# Electron stochastic acceleration in laboratory-produced kinetic turbulent plasmas

Dawei Yuan [1,2,18], Zhu Lei [3,4,5,18], Huigang Wei[1], Zhe Zhang [6,7,8], Jiayong Zhong [2,9], Yifei Li[6], Yongli Ping[9], Yihang Zhang[6], Yutong Li [6,7,8,10] ✉, Feilu Wang[1,11], Guiyun Liang[1,2], Bin Qiao [4,5,12] ✉, Changbo Fu [13], Huiya Liu[14], Panzheng Zhang[14], Jianqiang Zhu[14], Gang Zhao [1,11] ✉ & Jie Zhang [7,15,16,17] ✉

The origin of energetic charged particles in universe remains an unresolved issue. Astronomical observations combined with simulations have provided insights into particle acceleration mechanisms, including magnetic reconnection acceleration, shock acceleration, and stochastic acceleration. Recent experiments have also confirmed that electrons can be accelerated through processes such as magnetic reconnection and collisionless shock formation. However, laboratory identifying stochastic acceleration as a feasible mechanism is still a challenge, particularly in the creation of collision-free turbulent plasmas. Here, we present experimental results demonstrating kinetic turbulence with a typical spectrum $k^{-2.9}$ originating from Weibel instability. Energetic electrons exhibiting a power-law distribution are clearly observed. Simulations further reveal that thermal electrons undergo stochastic acceleration through collisions with multiple magnetic islands-like structures within the turbulent region. This study sheds light on a critical transition period during supernova explosion, where kinetic turbulences originating from Weibel instability emerge prior to collisionless shock formation. Our results suggest that electrons undergo stochastic acceleration during this transition phase.

The origin of energetic particles in the universe is still an open question. Observations suggest that these particles mainly come from various explosive astrophysical events[1–5]. However, the exact acceleration mechanisms responsible for them remain a topic of debate. Magnetic reconnection acceleration (MRA)[6,7], converting magnetic energy into energetic particles, is considered a key energy dissipation in space and astrophysical plasmas, such as solar flares[1,2], substorms in Earth's magnetotail[8], and gamma-ray bursts[5,9]. Diffusive shock acceleration (DSA)[10,11], confining particles multiple passing through the shock front, is a well-known model to explain the power-law cosmic-ray distributions. The stochastic acceleration (SA)[12] model associated with turbulent plasmas is also suggested to interpret the observed

nonthermal emission from a strongly magnetized environment (with large magnetic Reynolds numbers). Although several acceleration scenarios have been proposed, there still exist numerous uncertainties regarding individual mechanisms, for example, the onset of MR with multiscale[13], the injection problem of DSA[14], and so on.

Motivated by these astrophysical challenges, laboratory experiments with scaled-down versions provide a novel approach to studying them in detail[15,16]. Recent studies have focused on particle acceleration in space and astrophysical plasmas. For instance, magnetic reconnection experiments relevant to different-$\beta$ astrophysical environments (where $\beta$ is defined by thermal-to-magnetic pressure ratio) have successfully demonstrated that the electrons can be

accelerated by the parallel electric field[17] or by perpendicular reconnection electric field[18]. Shocks with parameters relevant to astrophysical ones are produced using laser-driven piston plasma expanding into ambient gas under an externally applied magnetic field, where the protons are energized by shock surfing acceleration[19]. Electron acceleration is achieved in collisionless shocks where thermal electrons gain energy by experiencing the first-order Fermi process[20]. However, direct demonstration of stochastic acceleration in the experiment is still challenging due to the difficulties in the creation of kinetic turbulent plasmas and the low acceleration efficiency (defined by $\Delta E/E$, the average fractional energy gain per collision between electrons and scatters). The Weibel instability (WI)[21] arising from the flow velocity or temperature anisotropy can generate the electromagnetic turbulence associated with astrophysical shock formation. Experiments have shown that the Weibel instability can be induced by a relativistic electron beam propagating into a target[22], as well as by non-relativistic interpenetrating plasma flows[23–25]. Theoretical studies have also shown that SA becomes effective in multiple magnetic islands[26,27] or strongly magnetized plasmas[12]. In this paper, we present our recent experimental results that the kinetic turbulence is successfully produced through the self-organized WI and the thermal electrons within the turbulent region are stochastically accelerated to relativistic energy with a power-law distribution.

## Results

### Experimental setup and plasma conditions

The experiment was performed at Shenguang-II (SG-II) laser facility[28] at the National Laboratory on High Power Lasers and Physics. The experimental layout is shown in Fig. 1 (More details are shown in the

"Methods" and Supplementary Fig. 4). Two interpenetrating plasma flows with high Mach number ($M \sim 6$) were created by laser ablating a pair of opposite LiD powder targets (lithium deuteride: $Z_i = 2$, $A_i = 4$) separated by $L \approx 3.2$ mm. Both expanding plasma flows interacted with each other near the midplane. The spatiotemporal evolution of interaction was probed by time-resolved optical diagnostics. The interferometer was used to measure the plasma density based on the plasma refractive index of $N \approx 1 - n_e/n_c$, where $n_e$ is the plasma density and $n_c$ is the critical density of the probe. The shadowgraph was used to measure the denser filaments. A modified Faraday rotation method was used to measure the topology and strength of the Weibel magnetic fields. A time-integrated X-ray pinhole camera (PHC) diagnosed the Coulomb collisions and electron magnetic spectrometers (EMSs)[29] measured the electrons. One freely expanding plasma flow at 0.5 ns (referring to the end of drive lasers) was well characterized with plasma density $n_e \sim (0.5\text{-}1) \times 10^{19}$ cm$^{-3}$ obtained from the two-dimensional interferometry, maximum temperature $T \sim 600$ eV derived from the adiabatic expansion theory basing on the density profile and flow velocity $V_{flow} \sim 1.5 \times 10^8$ cm s$^{-1}$ measured by time-resolved interferometry with streak camera (see Supplementary Fig. 6). Table 1 summarizes the typical parameters of interpenetrating plasma flows. The mean free path (MFP) for ion-ion and ion-electron collisions within the inter-flow is substantially larger than the system size. This enables ions to freely interpenetrate between plasma flows, conforming to the observations from the PHC data. The electrons formed a thermalized background due to large thermal velocity ($v_{the} \gg V_{flow}$). Despite that the ion-ion MFP within the intra-flow is significantly smaller than the target size [see Supplementary Table 1], it will not impede the development of the WI. Previous works[23,30,31] have shown that the WI can fully

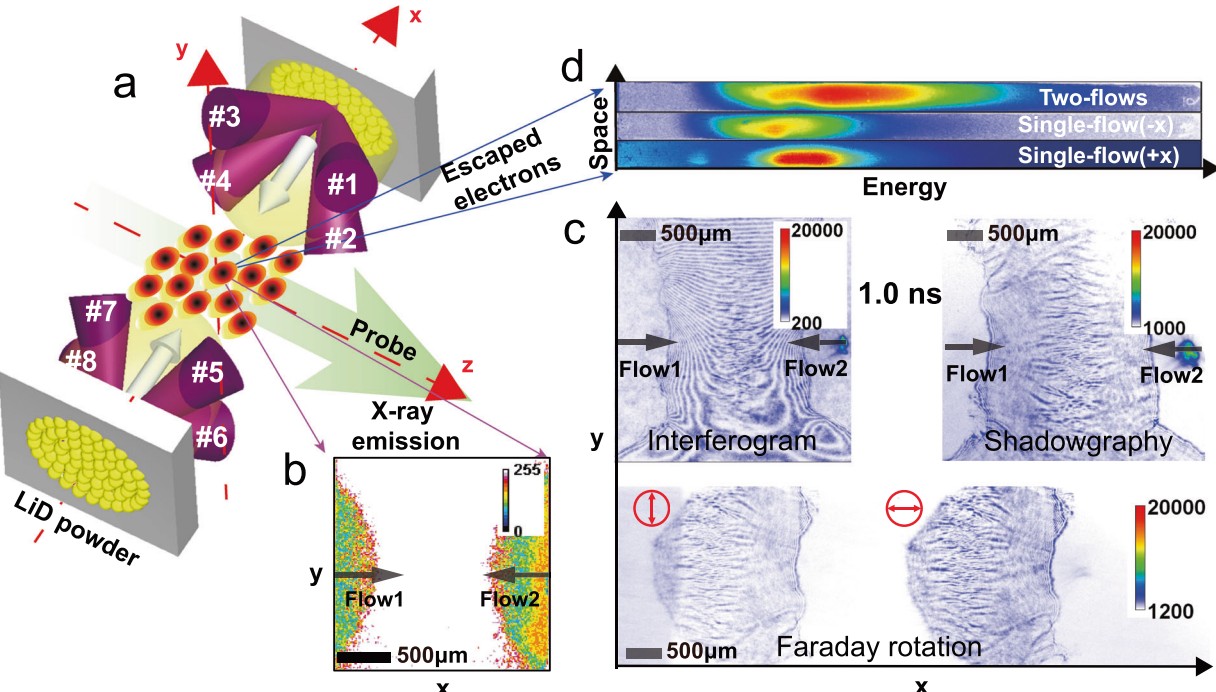

**Fig. 1 | Schematic of the experimental setup. a** Interpenetrating plasma flows (± X-direction) were produced by drive laser ablating the face-on LiD (lithium deuteride) powder embedded in the Copper holder (The powder size is randomly distributed in 20–100 μm.). The optical diagnostics, including interferometry, shadowgraphy, and Faraday rotation, viewed the interaction region in the z-direction (Details see Supplementary Fig. 4). The X-ray pinhole camera (PHC) placed below the equatorial plane was filtered with 50 μm allowing X-ray energy above 1 keV to pass. Three electron magnetic spectrometers (ESMs) positioned 17 cm away from the target center at different observation angles 30⁰, 50⁰, and 70⁰ relative to the plasma flow

(X-direction) measured the energetic electrons. **b** The detected X-ray self-emission is mainly from the bremsstrahlung caused by Coulomb collisions. The X-ray signal disappears in the interaction region, indicating that the Coulomb collisions between interpenetrating flows are weak and can be ignored. **c** Typical raw optical data obtained at 1.0 ns showing that Weibel filaments appeared at the interaction region. **d** The comparison of the time-integrated electron spectrum (raw data obtained by ESM at observation angle 70⁰) between interpenetrating flows and different single flows shows that many thermal electrons were accelerated to high-energy.

**Table 1 | Derived typical parameters for interpenetrating plasma flows**

| Parameters | Definition | Value |
|---|---|---|
| Ion-ion MFP, $\lambda_{i\text{-}i}$ (cm) | $670A_i^4(V_r[1000\,\text{km/s}])^4/(Z_i^4(2A_i)^2 n_i[10^{19}\,\text{cm}^{-3}]\ln\Lambda)$ | ~200 |
| Ion-electron MFP, $\lambda_{i\text{-}e}$ (cm) | $6\times10^{-3}A_i V_r[1000\,\text{km/s}]\,T_e^{3/2}(\text{eV})/((Z_i^2 n_e[10^{19}\,\text{cm}^{-3}]\ln\Lambda)$ | ~31 |
| Eletron-electron MFP, $\lambda_{e\text{-}e}$ (cm) | $0.5(T_e[0.5\,\text{keV}])^2/(n_e[10^{19}\,\text{cm}^{-3}]\ln\Lambda)$ | ~$10^{-3}$ |
| Electron inertial length, $d_e$ (μm) | $c/\omega_{pe}$ | ~1.7 |
| Ion inertial length, $d_i$ (μm) | $c/\omega_{pi}$ | ~100 |
| Electron thermal vel., $u_{the}$ (cm s$^{-1}$) | $4.2\times10^7 T_e^{1/2}$ | $1.0\times10^9$ |
| Ion thermal vel., $u_{thi}$ (cm s$^{-1}$) | $9.8\times10^5 A^{-1/2} T_i^{1/2}$ | $1.2\times10^7$ |
| Flow Mach number, $M$ | $V_{flow}/C_s$ | ~6 |
| Growth rate, $\Gamma$ (s$^{-1}$) | $0.1\times u/c\times\omega_{pi}$ | ~$1.5\times10^9$ s$^{-1}$ |

Derived parameters describing the experimental platform are based on the flow conditions, including the relative velocity $V_r = 2V_{flow} \approx 1.5\times10^8$ cm s$^{-1}$, average atomic number $A_i = 4$, electron density $n_e = Z_i n_i \approx 10^{19}$ cm$^{-3}$, the Coulomb logarithm $\ln\Lambda \sim 8.5$, plasma temperature $T = T_e = T_i \approx 600$ eV (corresponding the sound velocity $C_s = 2.6\times10^7$ cm s$^{-1}$), and the electron (ion) plasma frequency $\omega_{pe} = 1.8\times10^{14}$ s$^{-1}$ ($\omega_{pi} = 3\times10^{12}$ s$^{-1}$).

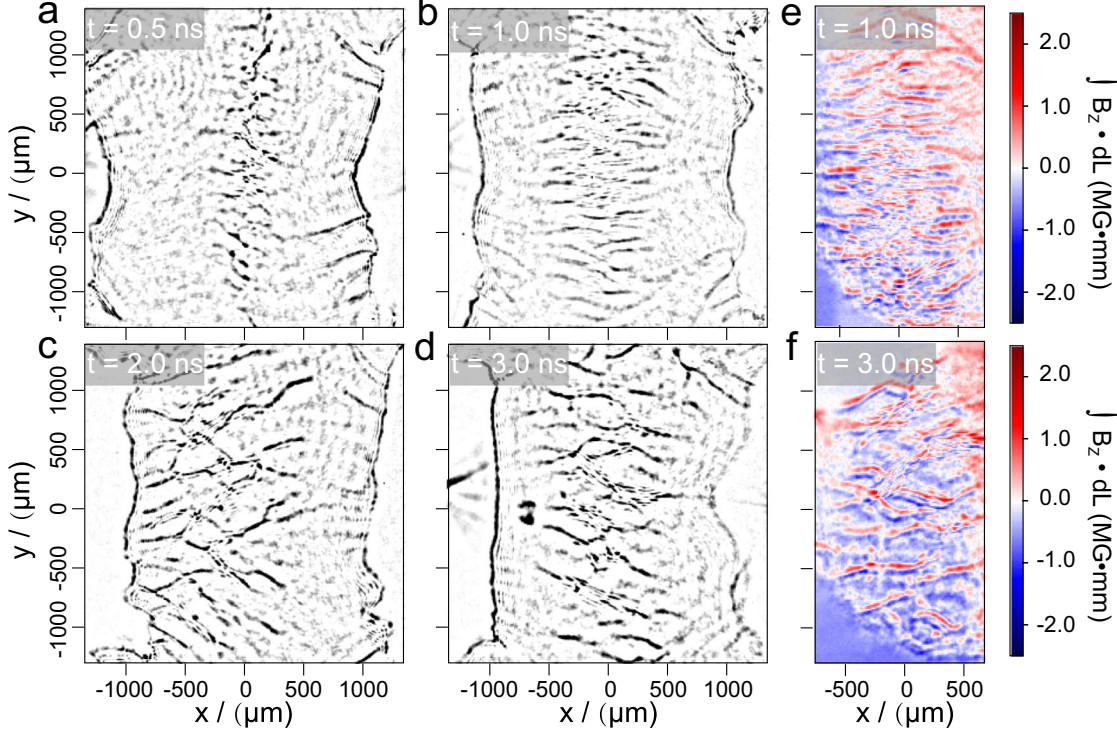

**Fig. 2 | Experiment demonstrating the evolution of Weibel instability.** Filaments in (**a**–**d**) measured using shadowgraphy represent the information of plasma density, and in (**e**, **f**) measured using the Faraday method represent the path-integrated magnetic field strength. **a**, **b** Linear phase (0.5 and 1.0 ns) of WI showing that locally ordered filaments form at the midplane and stretch along the flow direction with both flows' interpenetration. **c**, **d** Nonlinear phase (2.0 and 3.0 ns) of WI showing that filaments/currents coalesce each other and distort. **e**, **f** The typical distribution of transverse magnetic field path integral intensity ($\int B_z$dl) at 1 ns (**b**) and 3 ns (**d**). These images have been sharpened in postprocessing to emphasize the striations. (Raw images see Supplementary Fig. 5).

develop under the current experimental conditions. The condition of keeping ion velocity anisotropy provided an efficient driven mechanism for Ion-driven-Weibel-Instability. A fast-theoretical-linear growth rate $\Gamma \sim 1.5\times10^9$ s$^{-1}$ (more details see Supplementary information) showed WI quickly entering the nonlinear stage.

**The measurement of Weibel instability**

The measurement and corresponding simulation show that both interpenetrating flows last much longer than 15 ns, supporting our study of turbulence formation (see Supplementary Figs. 7 and 8). The representative snapshots of Weibel filaments and self-generated magnetic fields were obtained by optical diagnostics as illustrated in Fig. 2. In the linear stage (Fig. 2a, b), the regular periodical filaments initially develop from the interface and elongate along the flow direction ($\pm X$). The spacing between filaments is almost constant (comparable with the ion inertial length $\sim c/\omega_{pi}$), while the longitudinal length grows rapidly with a maximum length of about 900 μm ($\sim 9c/\omega_{pi}$). With the measurement time (0.5 ns and 1.0 ns), we can estimate that the linear growth time $t$ is about 0.5 ns. Consequently, a linear growth rate is obtained as $\Gamma \sim 1/t = 2\times10^9$ s$^{-1}$, close to the theoretical value as shown in Table I. After WI steps into the nonlinear phase (Fig. 2c, d), the dramatic changes mainly occur in the transverse direction where adjacent elongated filaments coalesce with each other, resulting in a decrease in the number of filaments and an increase in the transverse scale. Such deformation agrees well with the theoretical model[32] (see Supplementary Fig. 3c). The topology of the Weibel magnetic field shows a

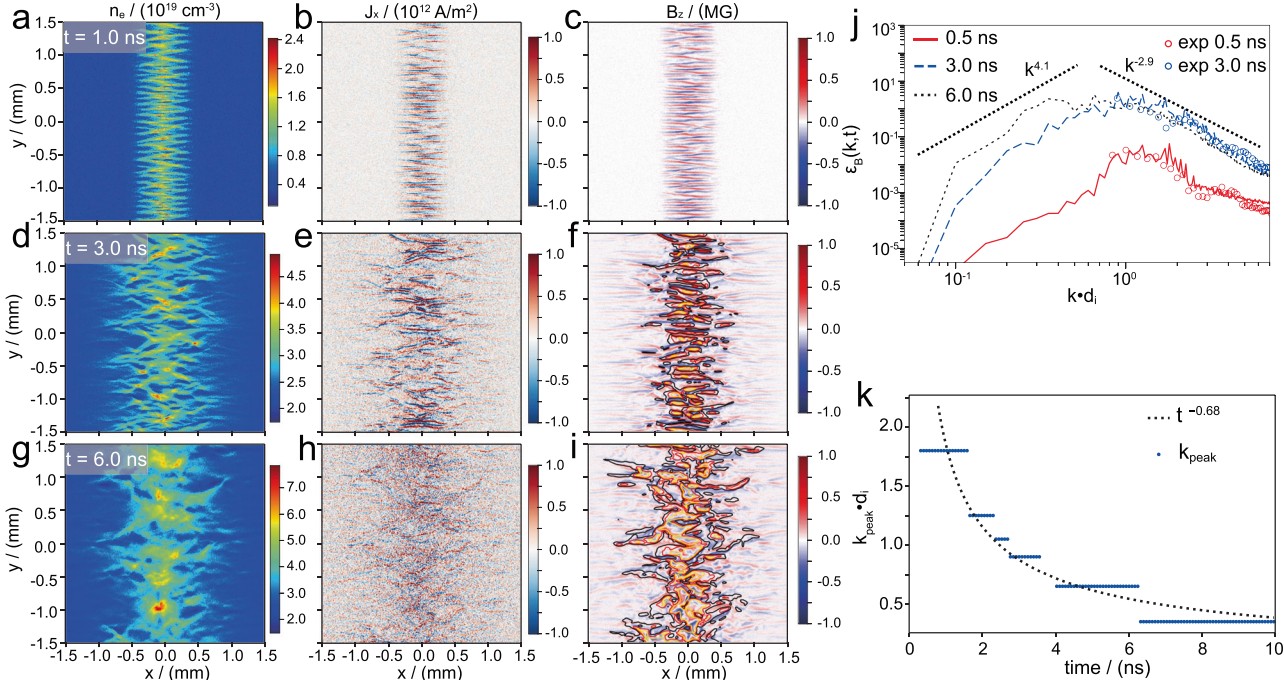

**Fig. 3 | Simulation revealing the Weibel instability features.** The plasma flows are injected into the simulation area from the left and right ends of the simulation box, and the number density and velocity of injection plasma are obtained via a radiation hydrodynamics simulation. Three distinct moments of interest are selected, 1.0 ns, 3.0 ns, and 6.0 ns, which correspond to linear and nonlinear stages of the evolution of WI, respectively. Panels (**a**–**c**) display the spatial distribution of the number density ($n_e$), x-directional current density ($J_x$), and z-directional magnetic field ($B_z$) at 1.0 ns. Correspondingly, panels (**d**–**f**) portray the corresponding data at 3.0 ns, and panels (**g**–**h**) are the corresponding data at 6.0 ns. The iso-contours in panel (**f**) and panel (**i**) represent the magnetic vector potential at 3.0 ns and 6.0 ns, and the black, red, and gold lines denote the iso-contours of the magnetic vector potential, indicating values of $2 \times 10^{10}$, $3 \times 10^{10}$, $4 \times 10^{10}$ Gs cm⁻¹ respectively. Panel (**j**) plots the magnetic field energy spectra $\varepsilon_B(k, t) = B(k)^2$ obtained by a fast Fourier transform at different times, and the red and blue circles represent the energy spectrum extracted from the reconstructed path-integrated magnetic field in experiments. Panel (**k**) shows the peak wave number $k_{peak}$ evolving with time.

similar evolution (Fig. 2e, f). The saturation strength of magnetic fields in Fig. 2f is about 0.75 MG obtained from the measured path-integrated magnetic field $\int B_z dl \approx 1.5$ MG mm, assuming the probe path length equals to the target width ∼ 2 mm.

## Simulations revealing the Weibel instability features

This self-organized deformation of WI observed in experiments is confirmed by scaled-down two-dimensional (2D) particle-in-cell (PIC) simulations, and further simulation details can be found in the Methods Section. Typical features of WI observed at different times in simulations are shown in Fig. 3. In linear phase, the magnetic fields grow exponentially (as depicted in Supplementary Fig. 3d) with initial magnetic perturbations $B_0 \sim 10^5$ G. The thermal ion Larmor radius, $r_{gthi} = m v_{thi} / q B_0 \sim 250$ μm is smaller than the transverse size ($L_y = 2$ mm), while drift ion Larmor radius, $r_{gdrift} = m V_{flow} / q B_0 \sim 3100$ μm is much larger than the interpenetrating depth ($L_x \sim 900$ μm). This leads to the motion of ions being characterized as Larmor precession, maintaining the velocity anisotropy to facilitate the development of WI. The ions in transverse directions (y) will be further converged by the magnetic perturbations, resulting in the generation of stronger currents. After that, WI steps into the nonlinear phase, where the magnetic fields continue to grow through the filamentary current coalescence process. In this phase, the drift-ion's Larmor motions become smaller as $r_{gdrift}$ (∼ 500 μm) comparable with $L_x$ when the magnetic field strength reaches 0.6 MG. This change will significantly disrupt the velocity anisotropy, leading to the saturation of Weibel magnetic fields. The maximum magnetic field is 0.75 MG, determined by the maximum current $I = J_x \times \pi R^2 \approx 125$ kA, approaching the Alfvén current limit ($I_A = 150$ kA)[33]. At time $t = 6$ ns, these strong elongated current filaments are distorted and fragmented into multiple magnetic islands, leading to the strongly turbulent plasma formation (Fig. 3i).

The average strength of magnetic fields in the turbulent region is about 0.6 MG, persisting for about 10 ns (see Fig. 4d–f). The magnetic vector potentials (**A**) in Fig. 3(f) and Fig. 3(i) are plotted by the iso-contour lines, which are intricately linked to the current density $J$ through the equation $\nabla^2 \mathbf{A} = -\mu_0 J$. We observe the emergence of distinctive magnetic island-like structures around 3 ns. Subsequently, at 6 ns, the disturbed region's overall scale expands, concomitant with the appearance of additional magnetic island-like structures, indicative of heightened plasma turbulence.

## Characteristics of turbulence and nonthermal electrons

As observations of fluctuations in astrophysical plasmas, turbulence is usually characterized by power-law energy spectra in density, velocity, and magnetic fields. Figure 3j shows the typical evolution of magnetic field energy spectra ($\varepsilon_B(k, t) = B(k)^2$), which is extracted from the reconstructed path-integrated magnetic field in experiments and the magnetic field strength in simulations. One can find that the WI starts growing in a broad $k$-spectrum with the peak wave number $k_{peak}$ (where the $\varepsilon_B(k, t)$ has the maximum value) appearing at $kd_i \sim 1$, indicating that the WI is driven by the ion velocity anisotropy. An inverse magnetic energy transfer from small to large scales is confirmed by the simulation, where the magnetic power increases rapidly on the ion skin depth ($d_i$), and the peak wave number $k_{peak}$ decreases with time. The typical evolution of peak wave number $k_{peak}$ obeying a power-law decay ($\propto t^{-0.68}$) is shown in Fig. 3k, following the self-consistent inverse cascade process[34]. Furthermore, in the large scale ($k < k_{peak}$), the power-law spectra evolve to a fitting of $k^{4.1}$, which tends to conform to the $k^4$ Batchelor spectrum, in agreement with the causality requirement $\nabla \cdot \mathbf{B} = 0$[35]. In the sub-inertial range ($k > k_{peak}$), the power-law spectra converges to $k^{-2.9}$, significantly deeper than the $k^{-2}$ from weak turbulence in magnetohydrodynamics (MHD)[36,37] and close to the $k^{-8/3}$ obtained in

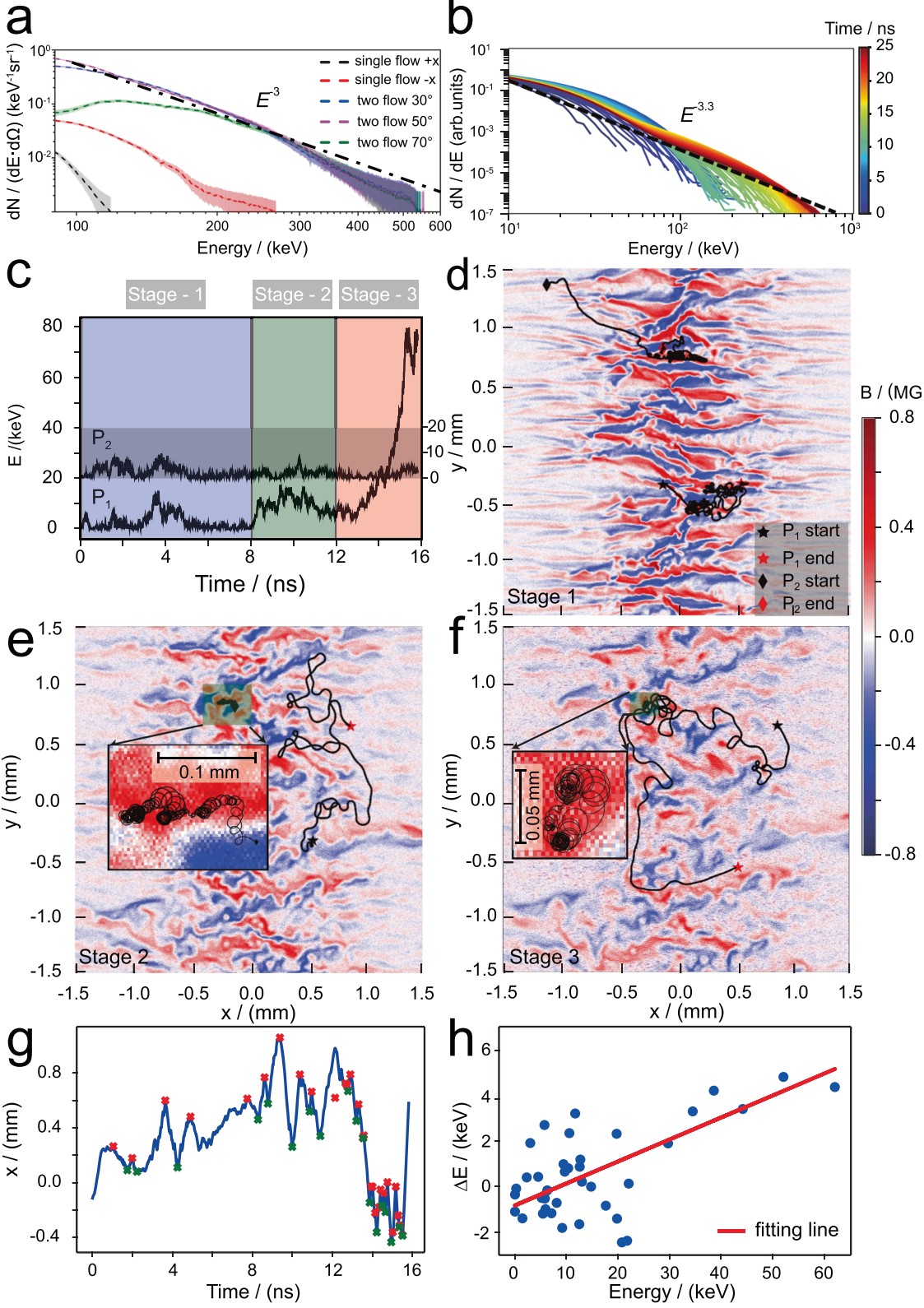

kinetic turbulent plasmas[38]. Although kinetic turbulence is also observed in the solar wind, the mechanism is different from that in our experiments. The turbulence in our experiments, which originates from Weibel instability, is highly compressible, and presumably characterized by fluctuation magnitudes ($dB$) comparable with the average magnetic field ($B$). In contrast, the kinetic-scale turbulence in the solar wind is dominated by kinetic Alfven waves and characterized by $dB/B \ll 1$[39].

Non-thermal electrons displaying similar power-law spectra with an index of $p = -3.0$ are unexpectedly obtained from different observation angles in the turbulent region (Fig. 4a and Supplementary Fig. 2). For comparison, two thermal electron backgrounds (< 0.3 MeV) with an approximate Maxwell distribution are obtained from two distinct single flows (see Supplementary Fig. 1). One background is from the flow propagation in the +$x$-direction, the other is from the flow

**Fig. 4 | The electron acceleration mechanism. a** The comparison of the time-integrated electron spectrum obtained from two-flow cases and single-flow cases. The energized electrons are observed in three directions showing similar power-law distributions with a spectral index of 3 (dotted black line), when the kinetic turbulence is formed. The systematically shaded region corresponds to the error bar of electron number ($\Delta n/n \sim 20\%$, where $\Delta n$ is the electron number error mainly caused by the sensitivity of the image plate). **b** Simulations show that the evolution of the electron spectrum in the turbulent region, where the background thermal electrons are accelerated to $\geq 100 k_B T_e$ leading to the formation of a power-law distribution with a spectral index of 3.3 (dotted black line). The color bar stands for the time evolution. **c** History of two typical selected electrons' kinetic energy as a function of time, one is an accelerated electron ($P_1$), and the other is a thermal electron ($P_2$). The acceleration process can be divided into three stages, and the dominant acceleration occurs in the last stage. Panels (**d**) to (**f**) show the trajectory of particles during different acceleration stages. The colormaps for (**d**) to (**f**) correspond to the distribution of magnetic fields at 4 ns, 10 ns, and 14 ns, respectively. Panel (**g**) represents the change of the x-coordinate of particle P1 over time, with the red and green crosses indicating points where its average velocity direction undergoes significant changes, representing the locations where reflection occurs. **h** The average energy gains experienced by electrons in each reflection in panel (**g**) are plotted.

propagation in the -x-direction. Recent experiments performed at the National Ignition Facility (NIF) have shown that non-thermal electrons with power-law distribution are produced by turbulent shock via 1[st] order Fermi acceleration[20]. However, in our experiments, the obtained turbulent region ($\sim 9c/\omega_{pi}$) is significantly lower than the theoretical prediction distance of supporting Weibel-mediated shock formation ($\sim 100c/\omega_{pi}$)[40,41]. Therefore, the shock acceleration mechanism can be ruled out in our findings. Our experimental data suggests the presence of another efficient electron acceleration mechanism. One promising mechanism associated with turbulence that we have identified is stochastic acceleration. The accelerated electrons display isotropic behavior, aligning well with our experimental observations. Through PIC simulations, we have found that kinetic turbulent plasmas have the capability to accelerate electrons from the thermal pool to energies nearing 600 keV within the experimental timescale, as depicted in Fig. 4b. Initially, the electron energy spectrum primarily displays a thermal pattern in the early stages, according to the PIC simulation results. However, as the turbulent plasma evolves, particularly in the later stages, approximately at $t \sim 20$ ns, the electron energy spectrum gradually transitions to a power law structure. Over time, the non-thermal electron distribution in the plasma region subjected to wave-particle interactions evolves towards a power-law energy tail, characterized by a spectral index of $p = -3.3$.

## Discussion

To capture the acceleration mechanism of non-thermal high-energy electrons, we sampled about $2 \times 10^5$ electrons in the turbulent region to record their motion trajectories during the simulation runs. Two typical selected particles are shown here (More particle trajectories see Supplementary Fig. 10). As depicted in Fig. 4c, $P_1$ represents the electrons ultimately accelerated to high energies, delineated into three acceleration stages according to their energy variations over time. Meanwhile, $P_2$ denotes thermal electrons exhibiting energy oscillations between 0 and 10 keV, and $P_2$ is also a typical representative of particles that have not been accelerated. From their trajectories, we find that $P_2$ initially ejects from the left side of the simulation box to the central part in the early stage. With the Weibel magnetic field enhancement, $P_2$ is bound in the magnetic island for gyrating motion due to the electron gyro-radius being much lower than the $L_M$ (as shown in the enlarged images in Fig. 4e, f so it fails to gain energy. For the $P_1$ electron, during stage 1, its trajectory is similar to that of $P_2$, which is trapped inside the magnetic island without significant change in energy. The acceleration first occurred during stage 2, when $P_1$ was lucky enough to leave the bound magnetic island to interact with the other one, as shown in Fig. 4e. Its energy increases from $0 \sim 10$ keV to $10 \sim 20$ keV. In stage 3, rapid acceleration occurs as the particle undergoes multiple collisions with magnetic islands, leading to a substantial energy boost from 20 keV to 80 keV, thereby transforming it into a suprathermal particle. In order to get the details of the electron acceleration, the change in the x-coordinate of particle $P_1$ over time is displayed in Fig. 4g. We calculate the variation resulting from reflections in the x-direction, which can be identified by the sign change of the electron velocity $v_x$[20]. The corresponding energy change for each reflection is displayed in Fig. 4f. One

can find that in the early stage where the electrons are at a low energy level (below 20 keV, conforming to the Hillas criterion[42] the maximum electron energy within the island is determined by the average strength of magnetic fields $B \sim 0.6$ MG and the characteristic scale of magnetic island-like structures $L_M \sim 300–400\ \mu m$), the electrons undergo a random acceleration process, i.e., energy-losing or energy-gaining, owing to the combination of head-on and head-tail collisions within the islands. In the later stages where the turbulent plasmas are fully developed, these electrons escaping from the islands obtain a net gain process. As illustrated in Fig. 4h, according to the simulation results, the average fractional energy gains experienced upon each reflection are $\Delta E/E \approx V_A/c \approx 0.1$ (where $V_A$ is the Alfven speed obtained by the average magnetic field strength of 0.6 MG). The acceleration efficiency becomes first-order in these multiple magnetic islands[22]. The overall process manifests a stochastic acceleration, where magnetic islands generated in turbulent regions act as scattering bodies.

Our experiments provide evidence for electron stochastic acceleration during collisions with magnetic islands in Weibel turbulent plasmas. In this process, thermal electrons gain energy in proportional to the Alfven speed ($V_A$). Similar dimensionless parameters between laboratory and supernova plasmas (as illustrated in Supplementary Table 2), allow our study to probe a transition phase of supernova explosion. This phase occurs when turbulence have generated before the formation of shock formation as the ejecta sweeping up the interstellar medium. Our experimental results provide a possibility that the process of electron stochastic acceleration associated with turbulence can take place in this transition period.

## Methods

### Laser conditions and optical diagnostics

Our experiment is performed at the SG-II laser facility, which consists of eight driven laser beams and one probe laser beam. The eight driven beams in a nanosecond square pulse, with a wavelength of 351 nm, are evenly divided into two bunches. Each bunch delivers a total energy of 1 kJ with an overlapped focus spot of $\sim 200–250\ \mu m$. Two laser bunches synchronously ablate the inner side of the facing targets to create the interpenetrating plasma flows. The probe laser used for the optical diagnostics works in two modes: short-pulse mode $\sim 80$ ps and long-pulse model $\sim 20$ ns, with a wavelength of 527 nm (shown as a green line). In the experiments, the parameters of single flow are measured with the probe working at long-pulse mode, and the evolution of WI is obtained from different shots with the probe working at short-pulse mode.

Optical diagnostics is widely used in laboratory astrophysics studies[17,43,44], here, we use it to measure the self-generated magnetic fields, plasma density, and filament structures. As shown in Supplementary Fig. 4, a probe beam propagating through the interested region is collected by an imaging system with a magnification of $\sim 2.5x$, and then is divided into three channels for Nomarski interferometer, shadowgraphy, and Faraday rotation. Such an arrangement makes the optical system share a magnification and is convenient for data processing, especially for Faraday rotation (Details see below).

As for the Faraday rotation method, the polarized probe with a rotation angle ($\Phi_r$) induced by the Weibel magnetic fields is divided by the Wollaston prism into two orthogonal polarization components, ordinary (O) and extraordinary (E) lights. Both modulated intensity distributions of O and E lights are ultimately obtained by a CCD with a large field of view. The measured intensity of each component is given by $r_O^l = S_O I_0 \sin^2(\theta + \phi_r)$ and $r_E^l = S_E I_0 \cos^2(\theta + \phi_r)$, where $I_O$ is the initial probe intensity, $\theta = 45^0$ is defined by setting a polarizer in front of the Wollaston prism, $S_O$ and $S_E$ are the transmission factors for each polarization. Here the $S_O$ and $S_E$ can be obtained from the measurement without magnetic fields case (before shooting), $I^O = S_O I_0 \sin^2(\theta)$ and $I^E = S_E I_0 \cos^2(\theta)$. Thus, the rotation angle $\Phi_r$ can be calculated by inversion of the intensity ratio of obtained images $r_O^l / r_E^l$. According to the Faraday effect, the rotation angle is determined by the magnetic field and electron density, expressed as (in Gaussian units) $\phi_r = \frac{\lambda^2 e^3}{2\pi e^m c^4} \int_0^l n_e(s) B_\parallel(s) ds = 2.62 \times 10^{-7} \lambda^2 \int_0^l n_e(s) B_Z(s) ds$, where $n_e$ is the electron density measured by the Nomarski interferometer, $\lambda = 5.27 \times 10^{-5}$ cm is the probe wavelength and the derived $B_z$ is the component of the Weibel magnetic fields parallel to the probe projection.

## Simulations and scaling laws

The fully-kinetic PIC simulations in this study are conducted in a two-dimensional (2D) xy plane using the "EPOCH" code[45]. The PIC simulations begin by utilizing the laser-ablated plasma profiles obtained from FLASH[46] prior to the interaction of the flows. The interaction commences 1.5 ns after the laser irradiation and is fully described kinetically. Due to the computational cost associated with kinetic simulations for the extensive temporal and spatial scales of the experiments, the majority of the simulations conducted were 2D3V. A self-similar transformation is proposed and implemented to establish a self-consistent connection between the PIC simulation and the radiation-magnetohydrodynamic (RMHD) simulation[47,48]. Initially, RMHD is employed to model the macroscopic states of plasmas, preceding the consideration of kinetic effects. Subsequently, PIC is utilized to simulate the ensuing kinetic processes. The injection of plasma velocity and number density in the subsequent stage is determined based on the results obtained from the RMHD simulation. Similar to the self-similarity principle of the magnetohydrodynamic equations[15,49], a set of free parameters is selected to scale down the plasma parameters obtained from the aforementioned RMHD simulation to a smaller scale that is suitable for kinetic PIC simulation. This scaling ensures that plasma properties, including $\beta$, $\beta_{ram}$, the Mach number, and the Alfvén Mach number, the ratio of ion cyclotron times between the two systems ($\omega_{ci,1}^{-1}/\omega_{ci,0}^{-1}$) et al., remain conserved between RMHD and PIC. The transformation equations are,

$$r_1 = \sqrt{f_m/f_n} r_0, \rho_1 = f_m f_n \rho_0, p_1 = f_n f_T p_0$$

$$V_1 = \sqrt{f_T/f_m} V_0, B_1 = \sqrt{f_n/f_T} B_0, t_1 = f_m/\sqrt{f_n/f_T} t_0 \quad (1)$$

where $r$, $\rho$, $p$, $V$, $B$, and $t$ respectively represent length, density, pressure, velocity, magnetic field, and time, and more details of this transformation can refer to our previous work[47].

We utilize the transformation methodology described in this section to perform kinetic simulations of plasma flow interaction. In order to strike a balance between computational efficiency and generality, we select the mean mass of ions in the PIC simulation to be $m_i = 165\,m_e$, where $m_e$ represents the mass of an electron, resulting in $f_m = 0.02$. In addition, we set the number density parameter $f_n = 1$ and the temperature parameter $f_T = 10$. These choices yield a time ratio of $t_I/t_0 \sim 6.3 \times 10^{-3}$, indicating that 6.3 ps in the PIC simulations corresponds to 1 ns in the RMHD simulations and experimental observations. Throughout the PIC simulations, the ion charge state $Z$ remains

consistent with that of the RMHD simulations. The simulation box dimensions after transformation are $L_x = 6$ mm and $L_y = 3$ mm, with plasma injection regions established on both sides. The density and velocity of the injected plasma are determined based on RMHD simulation results and experimental data, as illustrated in Supplementary Fig. 9. The simulation box is divided into grids of $2000 \times 1000$, corresponding to a resolution of $0.03 d_i$. We employ 100 macro-particles per species within each grid cell. To ensure the convergence of our simulation, we have verified that doubling the number of particles yields identical results to those presented in the images. Open boundary conditions are applied in all directions, and alternative sets of transformation parameters have minimal impact on the final outcomes. Similar to previous works[20,23], since the estimated average free path between particle collisions was much greater than the system scale, hence Coulomb collisions are not considered in our PIC simulations.

## Data availability

The data generated in this study have been deposited in the figshare database (https://doi.org/10.6084/m9.figshare.26036479). Source Data are also provided in this paper. Source data are provided with this paper.

## Code availability

The FLASH code used in this study is publicly available for download from https://flash.rochester.edu/site/flashcode. The EPOCH code in this study is publicly available for download from https://github.com/Warwick-Plasma/epoch.

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

## Acknowledgements

We thank the staff of the SG II laser facility and the target preparation for experimental cooperation. This work was supported by the National Key R&D Program of China (grant Nos. 2022YFA1603200, B.Q. and 2022YFA1603204, G.Y.L.), the Chinese Academy of Sciences Youth Interdisciplinary Team (JCTD-2022-05, D.W.Y.), the Youth Innovation Promotion Association of the Chinese Academy of Sciences (D.W.Y.), the National Natural Science Foundation of China (grants No. 11873061, D.W.Y., 12373027, G.Y.L., 12173058, H.G.W., 12325305, J.Y.Z.), and the Strategic Priority Research Program of the Chinese Academy of Sciences (grant No. XDA25030500, D.W.Y.).

## Author contributions

G. Zhao and J. Zhang. proposed and conceived the project of laboratory astrophysics. D.W.Y., H.G.W., J.Y.Z., Z.Z., Y.H.Z. and C.B.F. performed the experiment. D.W.Y., Z.L., H.G.W., H.Y.L., Z.Z., Y.L.P. and J.Y.Z. analyzed and processed the data. Z.L., Y.F.L., Z.Z. and B.Q. performed the simulations and interpreted the results. F.L.W. and G.Y.L. provided the additional theoretical support. D.W.Y., Z.L., Y.T.L., G.Zhao, and J.Zhang contributed to the writing of the manuscript. P.Z.Z. and J.Q.Z. were responsible for the operation of the SG-II laser facility.

## Competing interests

The authors declare no competing interests.

## Additional information

¹Key Laboratory of Optical Astronomy, National Astronomical Observatories, Chinese Academy of Sciences, Beijing, P. R. China. ²Institute of Frontiers in Astronomy and Astrophysics of Beijing Normal University, Beijing, P. R. China. ³Institute of Applied Physics and Computational Mathematics, Beijing, P. R. China. ⁴School of Physics, Peking University, Beijing, P. R. China. ⁵Center for Applied Physics and Technology, Peking University, Beijing, P. R. China. ⁶National Laboratory for Condensed Matter Physics, Institute of Physics, Chinese Academy of Sciences, Beijing, P. R. China. ⁷Collaborative Innovation Center of IFSA, Shanghai Jiao Tong University, Shanghai, P. R. China. ⁸Songshan Lake Materials Laboratory, Dongguan, Guangdong, P. R. China. ⁹Department of Astronomy, Beijing Normal University, Beijing, P. R. China. ¹⁰School of Physical Sciences, University of Chinese Academy of Sciences, Beijing, P. R. China. ¹¹School of Astronomy and Space Science, University of Chinese Academy of Sciences, Beijing, P. R. China. ¹²Frontiers Science Center for Nano-optoelectronic, Peking University, Beijing, P. R. China. ¹³Key Laboratory of Nuclear Physics and Ion-Beam Application (MoE), Institute of Modern Physics, Fudan University, Shanghai, P. R. China. ¹⁴National Laboratory on High Power Laser and Physics, Shanghai Institute of Optics and Fine Mechanics, Chinese Academy of Sciences, Shanghai, P. R. China. ¹⁵Tsung-Dao Lee Institute, Shanghai Jiao Tong University, Shanghai, P. R. China. ¹⁶Key Laboratory for Laser Plasmas (MoE), Shanghai Jiao Tong University, Shanghai, P. R. China. ¹⁷Department of Physics and Astronomy, Shanghai Jiao Tong University, Shanghai, P. R. China. ¹⁸These authors contributed equally: Dawei Yuan, Zhu Lei. ✉e-mail: ytli@iphy.ac.cn; bqiao@pku.edu.cn; gzhao@bao.ac.cn; jzhang1@sjtu.edu.cn

