## [Peer Review File · Nature Communications]

Electron stochastic acceleration in laboratory-produced kinetic turbulent plasmasREVIEWER COMMENTS

Reviewer #1 (Remarks to the Author):

The manuscript presents results demonstrating the evolution of ion Weibel turbulence produced in the head-on merging of two laser produced plasma streams. The authors make a convincing case for the presence of Weibel turbulence in the experiment -- with several lines of evidence pointing to the ion Weibel instability as the source of the observed turbulent fluctuations.

Accompanying PIC simulations additionally suggest that the observed spectrum of suprathermal electrons are the product of stochastic acceleration (SA) under the influence of magnetic island structures; which appear within the well-developed turbulence at late times.

While the manuscript is compelling, I have a number of concerns/questions.

1) The English needs some work in places. I have noted a number of typos throughout the manuscript and supplementary material [e.g., "Paeticle", instead of "Particle" in Supplementary Fig. 2].

2) There are a few issues with the Coulomb collisional mean free path (mfp) estimates.

2a) Firstly, the expression used in Table 1 is the mfp for a test particle slowing down in background of Maxwellian field particles. The expression is really only meaningful for mono-energetic beams (i.e., beams possessing very little thermal spread). In fact, for this experiment, the pre-merger ion thermal speed is about 1/10 of the flow velocity. Up to a numerical factor, the ion-ion mfp in the transverse direction (i.e., perpendicular to the stream velocity) is about $(0.1)^4$ times smaller than the slowing down mfp, and is actually below the system size. Since the Weibel filamentation instability arises from transverse deflections of particles due to magnetic perturbations in the plane normal to the beam direction, the thermal velocity mfp should be considered when assessing how collisional the plasma is.

2b) Secondly, the value of "u" used in the Table 1 should be the relative velocity (i.e., $2*u$), since the slowing down formula is for a test particle moving in a background at rest (in this case, the background is the other plasma stream).

2c) The mfp estimates should also include i-e collisions [as noted in Zakharov, High Energy Density Physics 8 (2012) 329-330, the i-e slowing down rate can actually be larger than the ion-ion one]. In this case, the i-e slowing down mfp is also large compared to the experimental scale, but this should be noted in the manuscript.

3) I noticed that the manuscript doesn't cite two previous works, which I believe it should. These are: S. Mondal et al., Proc. Natl. Acad. Sci. 109, 8011 (2012) and C. M. Huntington, et al., Nat. Phys. 11(2), 173-176 (2015). Both of these works present experimental results showing the development of the Weibel instability. Mondal et al. additionally includes an inference of the magnetic spectrum (similar to this manuscript).

4) The manuscript contains the passage: "Recently experimental results have reported that thermal electrons can be accelerated via a first-order Fermi process assisted by turbulent collisionless shock. However, this acceleration mechanism is ruled out in our experiment. First, the ion gyro-radius. . . is larger than the characteristic scale of magnetic island. . ."

It's true that the "drift" ion Larmor radius is larger, but the initial thermal ion Larmor radius is actually ~ 10 times smaller. I think the thermal Larmor radius should be considered too, since the magnetic fields only appear after the stream merger, and the directed kinetic energy of the streams is the source of the magnetic energy (hence, there should be a stagnation of the streams when the Weibel fields begin saturating).

5) Do the 2-D PIC simulations include Coulomb collisions?

6) The manuscript states: "At times later than that observed in our experiments (6ns). . .", implying that the experiment didn't last longer than 6 ns. However, Fig. 4c shows the P_1 and P_2 populations up to 16 ns. I presume that this is simulation data, but it's not clear from the text.

6a) Also, Fig. 4c shows that the P_1 population only becomes suprathermal well after $t = 6$ ns (Stage 3 @ ~ 14 ns). In fact, the P_1 population isn't even suprathermal at the time the experiment ended (~ 6 ns).

7) Given 6), if the acceleration doesn't even occur until much later than the experiment, how can it explain the purported power law in the observed electron energy spectrum?

7a) Fig. 4b also suggests an issue, since the PIC-derived power law only really develops around $t \sim 20$ ns, which is consistent with the bulk of the acceleration occurring in Stage 3. The lines around $t = 6$ ns don't really show a power law in the required window of energies.

7b) In fact, a plausible explanation for the two stream experimental spectrum in Fig. 4a, and the $t \sim 6$ ns PIC spectrum, is that they represent hotter, perturbed, Maxwellian distributions. This would make sense, as the merging of the streams would heat them (particularly if collisions aren't that negligible), and kinetic effects would then account for a deviation from a full Maxwellian distribution.

8) The self-similarity argument for translating the RMHD simulations to manageable, scaled-PIC simulations makes sense only if the underlying ideal MHD symmetry remains unbroken in the system.

8a) The authors note this by mentioning that viscosity, heat conduction, etc. break this symmetry, but that these transport effects shouldn't appear because the system wide Reynolds number, etc. are large.

8b) However, the authors experimentally infer that the magnetic correlation length becomes smaller than the system size, implying that the fluid closure scheme underlying ideal MHD is in question for the system at the relevant scales. For this reason, we may not expect a one-to-one correspondence between laboratory-scale RMHD simulations of the problem, and kinetic simulations at a reduced scale.

In summary, the results showing the structure of the magnetic turbulence -- which is believably the result of the ion Weibel instability (given the observed current filament evolution, and the correspondence between the theoretical Weibel growth rate and the experimentally inferred one) -- are compelling.

However, the core result of the paper is built upon a shaky foundation. I am not convinced that the semblance of a power law in the observed electron energy spectrum points to SA as the only plausible explanation.

This work should be published, but it may not be appropriate for Nature Communications. While the observed magnetic and electron spectra offer a valuable glimpse into the dynamics of Weibel turbulence, the results do not go far beyond previous works (e.g., Mondal et al.). If a more convincing case could be made for the SA mechanism, I may change my mind.

Reviewer #2 (Remarks to the Author):

Report on NCOMMS-23-56222, "Electron stochastic acceleration in laboratory-produced kinetic

turbulent plasmas" by Yuan et al.

This manuscript reports new results from laboratory experiments on electron acceleration by kinetic plasma turbulence generated by Weibel instability. Particle acceleration has been an outstanding problem for plasma astrophysics for a long time and most of the relevant research has been performed either theoretically, numerically, or observationally. The reported laboratory results provide a much-needed test of some of the proposed acceleration mechanisms. Previous experiments were performed on a larger laser facility (NIF) which generated Weibel instability-mediated collisionless shocks, which led electron acceleration (Ref. 20). The reported experiments here were performed at a smaller laser facility (Shenguang II), and the plasmas did not have enough sizes (in the unit of ion gyro-radius or ion skin depth) to form a complete shock structure, but similar electron acceleration was detected (Fig.4a, as compared to Fig.4a of Ref.20). Therefore, the observed electron acceleration should be only by the plasma turbulence mediated by Weibel instability but without shocks. The measurements of this plasma turbulence shown in Fig.2 and Fig.1a are of high quality, which have been analyzed for their spectra shown in Fig.3k. These results are sufficiently important and should be published in Nature Communications.

However, I would like to offer the following suggestions for authors to improve their manuscript.

1. The author should highlight the fact that similar electron acceleration was observed without collisionless shocks. It is difficult for readers to find out from the current version of the manuscript about this important difference unless they read Ref.20 and compared. Please note that this does not reduce the importance of the results reported in Ref.20 nor by this manuscript, and there is an important physics insight (that similar electron acceleration occurs with and without shocks) by comparing both experiments.

2. It is important to have control cases with only single flow as shown in Fig.4a and in Fig.1c. Please indicate which single flow between a flow from the negative x direction and a flow from the positive x direction. Please also show results from the single flow from another direction of X-axis. Since the ESM has an angle of 70 degrees from the X-axis, these two measurements are unlikely the same so showing both single flow cases are important.

3. According to numerical simulations, plasma turbulence should be much stronger at $t=6\text{ns}$ (Fig.3), especially including the so-called "magnetic islands", and electron energetic tail takes a long time to develop, as long as 15-25ns (Fig.4b, Fig.4c), but experimental data are shown only until $t=3\text{ns}$. Are there experimental data supporting this important development?

4. V_{flow} is used in line 137 while u is used in line 141. Please check.

5. Please cite some references on Weibel instability-mediated turbulence – experimentally and numerically.

Reviewer #3 (Remarks to the Author):

The manuscript "Electron stochastic acceleration in laboratory-produced kinetic turbulent plasmas" presents results of laboratory experiments on laser-produced plasmas that demonstrate electron acceleration associated with development of Weibel instability in interpenetrating flows. Based mainly on results of supporting computer simulations, it is argued that acceleration is stochastic and is associated with reflection by magnetic structures.

Overall, I believe that the manuscript contains results that merit publication in Nature Communications in the sense that they are sufficiently novel and are of interest to a broad community of researchers. However, I also feel that major improvements are needed before the manuscript

becomes suitable for publication.

Specifically:

- 1) The manuscript, especially abstract and the introduction, could benefit from careful proof-reading and editing. Many of the sentences are poorly constructed. I point out a few of those below, but it's not the job of a reviewer to proof-read and edit the text.
- 2) Out of the 4 Figures in the main section, I believe 3 need improvements (summarized below)
- 3) The manuscript places a great deal of attention on the stochastic nature of the observed electron acceleration. While this might be true, the relevant presentation focuses on analysis of 2 electron trajectories in the simulation. Some statistical evidence is required to demonstrate that stochastic acceleration by turbulence is indeed the dominant mechanism.

More detailed comments, in order of appearance in the text:

- 1) Line 62: "magnetic reconnection acceleration". Please be more specific. There has been a great deal of work (and some controversy) on the exact mechanisms accelerating particles in magnetic reconnection regions. Some examples include acceleration by parallel electric field, stochastic acceleration by contracting magnetic island, curvature drifts, etc
- 2) In the same sentence: the first part reads "astronomical observations...indicate..."; while the second part says "...it would be difficult to identify from observations". This needs some re-phrasing. Perhaps one can say that observations indicate acceleration takes place, while theory and simulations identified various mechanism mentioned in this sentence.
- 3) Line 70 and throughout the manuscript: in what sense is "self-organized Weibel instability" being used? It might be appropriate to say "turbulence originating from Weibel instability" or something along those lines.
- 4) Line 75: I fail to see how these results imply that stochastic acceleration may be an efficient mechanism in various environments. That would depend on the type of turbulence being produced, etc; Here and in many other places throughout the text I get the impression that the rigor of presentation is being sacrificed to make the results more broadly appealing. I think this is a mistake.
- 5) I also don't see how the manuscript addresses the issue of particle injection into other acceleration processes. That scenario has been proposed before and in my opinion the manuscript does not add anything specific apart from showing that Weibel instability can indeed accelerate particles as expected. The latter is a very good experimental result, but it does not add to our understanding of the pre-acceleration problem.
- 6) Lines 102, 105: these sentences need editing (along with many others in the manuscript)
- 7) Line 111: "externally applied", or "external"? Also, I would not say "astrophysical magnetized shocks". At best, one can say that such shocks have parameters relevant to astrophysics, but even that is highly debatable (e.g. system sizes in all experiments are tiny even in normalized units)
- 8) Lines 116 & 117: perhaps acceleration efficiency needs an explicit definition here.
- 9) Line 131-132 or somewhere in the description of Fig 1: I think it would be good to bring here some information from the "Methods" section on what each individual diagnostic measures.
- 10) Table 1: it might be good to also include electron mean free path or collisional frequency.
- 11) Line 143: I am assuming that the growth rate is a theoretical value: if yes, please specify that. In the method section, the description of how the growth rate was obtained is limited to simply stating that it comes from an electromagnetic solver. Please describe the solver being used (what equations are being solved?), parameters used in the calculation, and the equilibrium assumed.
- 12) Fig 1:
 - a. Color bars in panel a) are unreadable, unless one zooms in. It is possible to that on screen, but it would not be possible in print.
 - b. The vertical axis in panel c) needs a label.
 - c. Line 153 in the caption: "No signal appearing in the central region indicates...": - this needs an explanation.
 - d. I would also suggest labeling the left-most schematic (i.e. labels would go a-d instead of having one

un-labeled sketch)

13) Lines 159 -160 and in other places “the space ... almost keeps constant” – is it better to say “the spacing ... is almost constant”?

14) Lines 161-163: it’s not exactly clear how the linear stage is identified. Is the time-resolution of this diagnostic 0.5 ns? Or are these two snapshots chosen as the most representative? If the former, the presented growth rate is a lower bound at best, since the growth time can be much shorter. If the second, perhaps one can show evolution of an average magnetic field magnitude?

15) Fig. 2: please label axes. I would also add in the caption some information on what panels a) to d) represent (density? current?)

16) Lines 179-180: “scaled-down two dimensional ...” is probably better than “scaling-down two dimensional”

17) Line 190: since the subsequent discussion appeals to magnetic islands, it might be useful to show the magnetic field topology (e.g., iso-contours of vector potential in 2D or something equivalent).

18) Lines 219-223: I think the assumption that all kinetic-scale plasma turbulence is characterized by the same spectrum is not universally accepted, so appealing to it may be ill-advised. More broadly speaking, I would also say that the turbulence in these experiments is very different from the solar wind. Here, it is strongly compressible, originating from Weibel instability, and is presumably characterized by fluctuation magnitude δB comparable with the mean field B_0 . In the solar wind, the kinetic-scale turbulence is believed to be dominated by kinetic Alfvén fluctuations, is characterized by very weak fluctuation amplitudes $\delta B/B_0 \ll 1$ (at kinetic scales), is nearly perpendicular, etc.

19) Discussion in lines 233-234: perhaps it is also good to mention (and describe) relevant simulation results here?

20) Fig 4: I have lots of comments here:

a. Panel a): one can argue that single flow spectra are also characterized by the same power-law in the range 200-300 eV, even if amplitude is lower.

b. Panel b): why not use the same energy units as in panel a)?

c. Panel b): could you comment on why the time to achieve the power-law spectra is substantially longer in the simulation compared to the experiment?

d. Panels d)-f) are incomprehensible for two reasons: a) use of color (I am a colorblind person and cannot see the trajectories); b) they are too small; I believe it is common practice to try to make figures accessible.

21) More broad comment about results summarized in Fig. 4: all said and done, this is analysis of two electrons that shows acceleration events that are characterized by large jumps in position. If one is to make a point that dominant acceleration mechanism is stochastic, some statistical analysis of many electrons is needed. I would probably suggest focusing this analysis on energetics of individual events.

22) Lines 286: “the average energy gain...”: is this a result of an analysis of the simulation or an assumption? If the former, please describe the analysis.

23) The same comment about line 288: “the acceleration efficiency becomes first order in these multiple magnetic islands”. Does this statement apply to this particular simulation, or is it a generic statement based on Ref. 22?

24) Line 292: “although both systems ...” does “both” here refer to laboratory and astrophysical plasmas? There is a wide variety of systems in laboratory and especially in astrophysics that are characterized by vastly different parameters, acceleration mechanisms, etc. I think this statement is too generic.

25) Line 308: As I already stated earlier, I don’t think it’s a good idea to equate Weibel turbulence to solar wind turbulence. I don’t believe Weibel turbulence teaches us anything about dissipation in the solar wind and warm interstellar medium, and likely in other systems listed there.

Methods section:

26) I think the description of the scaling assumption needs to be simplified. It is long and confusing. The kinetic equations are clearly not self-similar under the specified scaling. One can simply state, as is the common practice in the field, which dimensionless parameters are kept the same between simulations and real system and which ones are changed.

27) Line 489 "the mean charge state remains consistent"; Does this imply that the charge state of individual ions can change in the simulation?

28) Please provide complete information necessary to reproduce the simulation: parameters of the injected plasma (flow speed, temperature, density); how the simulation is initialized; how the injection is done. The goal should be ensuring that the simulation can be reproduced by an interested researcher. It is not possible based on the information provided. As I mentioned earlier, the same comment applies to the description of the linear growth rate calculation.

Response Letter

Manuscript NCOMMS--23-56222

Response to Reviewer's comments

"Electron stochastic acceleration in laboratory-produced kinetic turbulent plasmas" By D.W. Yuan, Z. Lei, et.al.

We thank all the reviewers for their careful reading of our manuscript and providing insightful comments. Their constructive comments have greatly enhanced the quality of our manuscript that presenting the important aspects of stochastic acceleration in kinetic turbulent plasmas.

Based on the reviewers' comments and suggestions, we have made significant modifications in the revised paper. Additional experimental evidences and discussions have been included in the text to further support the concept of electron stochastic acceleration. We also carefully considered and incorporated each suggestion and comment made by the reviewers. Each point raised by the reviewers has been either addressed in the revised manuscript and supplementary information, and noted here, or addressed in detailed responses to the reviewers. In the following sections, we present the reviewer's comments in blue italics, our reply is in black plain type, and modifications in the revised manuscript (list in this reply) are highlighted in red.

All changes in the revised manuscript and supplementary information are summarized as follows:

List of Changes

1. In response to Comment 2-2 of Reviewer #1, we have estimated the thermal ion collisions within one flow and added a short paragraph to consider collisional effect on the Weibel instability in our manuscript, see Line **149-156** and **Table 1**, and **Section I** in the Supplementary Information.
2. In response to Comment 2-4 of Reviewer #1, we have added the i-e collision mean free path, see Line **149-152** and **Table 1**, and **Section I** in the Supplementary Information.
3. In response to Comment 3 of Reviewer #1, we have cited the previous works in our introduction, see Line **122-124**.
4. In response to Comment 4 of Reviewer #1, we have calculated the thermal ion Larmor radius and added discussions, see Line **203-211**, **213-215**.
5. In response to Comment 5 of Reviewer #1, we have added the information of the Coulomb collisions in the Methods section, see Line **506-508**.
6. In response to Comment 6 of Reviewer #1, we have added some sentences to explain the experimental time-scale in the revised manuscript and added relevant experimental data in the supplementary information, see Line **173-175**, and **Section VII** in the Supplementary Information.
7. In response to Comment 6-1 of Reviewer #1, we have modified the related descriptions, see Line **319-321**.
8. In response to Comment 7-1 of Reviewer #1, we have added the descriptions in our revised manuscript, see Line **276-287**.
9. In response to Comment 8 of Reviewer #1, we have modified the expression for scaling-law of the PIC methods, see Line **481-484**.
10. In response to Comment 9 and Comment 10 of Reviewer #1, we added important experimental data and explanations to support our findings of electron stochastic acceleration, see modified **Fig.1**, **Fig.4(a)**, Line **276-278**, **Section III** and **VIII** in the Supplementary Information.
11. In response to Comment 2-1 of Reviewer #2, we have added some sentences to highlight the recent experiments, see Line **269-275**.
12. In response to Comment 2-2 of Reviewer #2, we have added the experimental data and some sentences to explain the background of thermal electrons in the revised manuscript, see modified **Fig.1**, **Fig. 4(a)**, Line **266-269**, and **Section II** in the Supplementary Information.
13. In response to Comments 5 of Reviewer #3, we have modified the abstract, see Line **61-67**.
14. In response to Comments 7 of Reviewer #3, we have edited the Abstract, see Line **75-79**.
15. In response to Comment 9 of Reviewer #3, we have modified the sentences in our revised manuscript, see Line **101-106**.
16. In response to Comment 10 of Reviewer #3, we have modified the sentences in our revised manuscript, see Line **111-113**.
17. In response to Comment 12 of Reviewer #3, we have added the information of the

diagnostic, see Line **137-141** and more details can be found in **Section VI** of the Supplementary Information.

18. In response to Comment 14 of Reviewer #3, we have added the contexts of the theoretical growth rate in **Section IV** in Supplementary Information.
19. In response to Comment 15 of Reviewer #3, we have modified the **Fig.1** in our revised manuscript and the related caption, see Line **168-170**.
20. In response to Comment 18 of Reviewer #3, we have modified the **Fig.2** in our revised manuscript and the related caption, see Line **193-195**.
21. In response to Comment 20 of Reviewer #3, we have modified the **Fig.3** in our revised manuscript and the related discussion, see Line **221-227, 235-237**.
22. In response to Comment 21 of Reviewer #3, we have added a short discussion, see Line **257-263**.
23. In response to Comment 23 of Reviewer #3, we have modified the **Fig. 4** and the related caption in our revised manuscript, see Line **289-292**.
24. In response to Comment 24 of Reviewer #3, we have modified the related descriptions in our revised manuscript, see Line **306-310**.
25. In response to Comment 25 of Reviewer #3, we have added the description of the analysis, see Line **333-336**.
26. In response to Comment 27 of Reviewer #3, we have modified the related descriptions, see Line **341-347** and **Section V** in the Supplementary Information.
27. In response to Comment 29 of Reviewer #3, the scaling laws have been simply described in the Methods, see Line **486-493**.
28. In response to Comment 30 of Reviewer #3, we have added the contexts of the theoretical growth rate in **Section IV** in Supplementary Information.

I. Responses to Reviewer #1:

(1) *"The manuscript presents results demonstrating the evolution of ion Weibel turbulence produced in the head-on merging of two laser produced plasma streams. The authors make a convincing case for the presence of Weibel turbulence in the experiment -- with several lines of evidence pointing to the ion Weibel instability as the source of the observed turbulent fluctuations.*

Accompanying PIC simulations additionally suggest that the observed spectrum of suprathermal electrons are the product of stochastic acceleration (SA) under the influence of magnetic island structures; which appear within the well-developed turbulence at late times."

We thank the reviewer's encouraging comments and positive assessment of our evidence supporting the presence of the ion Weibel turbulence in our experiments. The feedbacks are greatly appreciated and will help us further improve our research.

(2) *"While the manuscript is compelling, I have a number of concerns/questions."*

Thanks for these valuable feedbacks. We have thoroughly read the comments. The necessary improvement and corrections have been implemented in the revised version of the manuscript.

(2-1) *"The English needs some work in places. I have noted a number of typos throughout the manuscript and supplementary material [e.g., "Paeticle", instead of "Particle" in Supplementary Fig. 2]."*

Thank you very much. We have corrected these spelling errors.

(2-2) *"There a few issues with the Coulomb collisional mean free path (mfp) estimates."*

"Firstly, the expression used in Table 1 is the mfp for a test particle slowing down in background of Maxwellian field particles. The expression is really only meaningful for mono-energetic beams (i.e., beams possessing very little thermal spread). In fact, for this experiment, the pre-merger ion thermal speed is about 1/10 of the flow velocity. Up to a numerical factor, the ion-ion mfp in the transverse direction (i.e., perpendicular to the stream velocity) is about $(0.1)^4$ times smaller than the slowing down mfp, and is actually below the system size. Since the Weibel filamentation instability arises from transverse deflections of particles due to magnetic perturbations in the plane normal to the beam direction, the thermal velocity mfp should be considered when assessing how collisional the plasma is."

We greatly appreciate the reviewer's insightful comment. It is valuable to consider the mean free path between thermal ions within one plasma flow, also known as intra-flow collisions. The ion-ion mean free path for intra-flow collisions can be calculated as [Nat. Phys. 16, 916 (2020)], $\lambda_{i^*i^*}[\text{cm}] \sim 0.5(T_e[0.5\text{keV}])^2/(Z^4n_i[10^{19}\text{cm}^{-3}]\ln\Lambda)$, mainly

determined by the plasma temperature. It is due to the fact that the longitudinal propagation velocity is primarily the flow velocity while the transverse expanding velocity is the thermal velocity. For our experimental conditions with a flow temperature of $T_e = T_i = 600$ eV, ion density $n_i = n_e/Z$, Coulomb logarithm $\ln\Lambda = 8.5$, we find $\lambda_{i^*-i^*} \sim 0.01$ cm. For completeness, we also estimated the mean free path for electron-electron and electron-ion within one plasma flow as 0.001 cm and 0.05 cm, respectively. It indicates that collisions (i-i, i-e, etc) within in flow are frequent and thus single flow behaves hydrodynamically. A comparison of Coulomb collisions has been summarized in Supplementary Table 1.

Furthermore, after estimating the thermal ion Larmor radius raised by the reviewer as below, one can find that these values relevant to the ions (ion-ion collisions within one flow, ion inertial length, the filament spacing) are almost similar. It allows the Weibel magnetic perturbations to effectively deflect thermal particles and generate stronger current formation. This positive feedback ensures the full development of the Weibel instability.

Previous experimental (Nat. Phys. 11, 173 (2015); Phys. Rev. Lett. 111, 225002 (2013)) and theoretical works (Phys. Plasmas 21,032701 (2014)) have also shown that the development of Weibel instability in current experimental conditions closely resembles the purely collisionless case. Intra-flow collisions are not expected to significantly impact the overall evolution.

To clarify this physics picture of Weibel instability, we have added a short paragraph in the main text that reads [see item 1 in the “List of Changes”]:

“The mean free path (MFP) for ion-ion and ion-electron collisions within the inter-flow is substantially larger than the system size. This enables ions to freely interpenetrate between plasma flows, conforming the observations from the PHC data. The electrons formed a thermalized background due to large thermal velocity ($U_{the} \gg V_{flow}$). Despite that the ion-ion MFP within the intra-flow in the transverse direction is significantly smaller than the target size [see Supplementary Table 1], it will not impede the development of the Weibel instability. Previous works [Nat. Phys. 11, 173 (2015); Phys. Rev. Lett. 111, 225002 (2013); Phys. Plasmas 21,032701 (2014)] have shown that the Weibel instability can fully develop under current experimental conditions.”

(2-3) *“Secondly, the value of “u” used in the Table 1 should be the relative velocity (i.e., $2*u$), since the slowing down formula is for a test particle moving in a background at rest (in this case, the background is the other plasma stream).”*

We thank the reviewer for pointing this out. The relative velocity (V_r) between the two interpenetrating flows (V_{flow}) is twice the flow velocity. The ion-ion mean free path has been recalculated to be 200 cm, which is significantly larger than our target size of approximately 0.3 cm. We have corrected this important value in our revised manuscript.

(2-4) "The mfp estimates should also include i-e collisions [as noted in Zakharov, High Energy Density Physics 8 (2012) 329-330, the i-e slowing down rate can actually be larger than the ion-ion one]. In this case, the i-e slowing down mfp is also large compared to the experimental scale, but this should be noted in the manuscript."

We have estimated the electron-ion mean free path in the interpenetrating flows, which can be expressed as [HEDP 8, 329 (2012)],

$$\lambda_{i-e}^{inter}[\text{cm}] \sim 6 \times 10^{-3} A_i V_i [1000 \text{ km/s}] T_e^{3/2} (\text{eV}) / ((Z^2 n_e [10^{19} \text{ cm}^{-3}]) \ln \Lambda).$$

Taking our experimental conditions with a flow temperature of $T_e = T_i = 600$ eV, relative velocity $V_r = 2V_{flow} = 3000$ km s⁻¹, ion density $n_i = n_e/Z$, Coulomb logarithm $\ln \Lambda = 8.5$, we find $\lambda_{i-e} \sim 31$ cm, also larger than the target size, suggesting that collisions within inter-flow between electrons and ions can also be ignored. All the parameters about collisions in interpenetrating flows has been included in the revised manuscript and Supplementary information [see item 2 in the "List of Changes"].

(3) "I noticed that the manuscript doesn't cite two previous works, which I believe it should. These are: S. Mondal et al., Proc. Natl. Acad. Sci. 109, 8011 (2012) and C. M. Huntington, et al., Nat. Phys. 11(2), 173–176 (2015). Both of these works present experimental results showing the development of the Weibel instability. Mondal et al. additionally includes an inference of the magnetic spectrum (similar to this manuscript)."

Thank you for bringing these relevant prior works to our attention. We have referenced these two representative works and cited them appropriately. Some highlighted comments are also made in our literature section. Mondal *et al* firstly reported turbulent magnetic fields arising from the Weibel instability driven by the relativistic electrons. Huntington *et al* presented the development of Weibel instability associated with astrophysical shocks in interpenetrating flows. When discussing the turbulent magnetic spectrum, we have also cited the work of Mondal *et al*.

To highlight these representative works, we have added a short paragraph in the text that reads [see item 3 in the "List of Changes"]:

"Experiments have showed that the Weibel instability can be induced by a relativistic electron beam propagating into a target, as well as by the non-relativistic interpenetrating plasma flows."

(4) "The manuscript contains the passage: "Recently experimental results have reported that thermal electrons can be accelerated via a first-order Fermi process assisted by turbulent collisionless shock. However, this acceleration mechanism is ruled out in our experiment. First, the ion gyro-radius. . . is larger than the characteristic

scale of magnetic island. . ."

It's true that the "drift" ion Larmor radius is larger, but the initial thermal ion Larmor radius is actually ~10 times smaller. I think the thermal Larmor radius should be considered too, since the magnetic fields only appear after the stream merger, and the directed kinetic energy of the streams is the source of the magnetic energy (hence, there should be a stagnation of the streams when the Weibel fields begin saturating)."

Again, we greatly appreciate the reviewer's insightful comment. It is important to consider two types of ion Larmor radius in this context. One is the drift-ion Larmor radius, and the other one is the thermal-ion Larmor radius. In the initial stage of Weibel instability, the thermal-ion Larmor radius $r_{gthi} = mu_{thi}/qB_0 \sim 240 \mu\text{m}$ is much smaller than the plasma size, indicating that the ions are easily deflected by the Weibel magnetic perturbations and generate stronger current to support the development of Weibel instability. While the drift-ion Larmor radius $r_{gdrift} = mV_{flow}/qB_0 \sim 3000 \mu\text{m}$ is much larger than the plasma size, suggesting that the motion of ions can be regarded as Larmor precession keeping ion drift velocity anisotropy. When the Weibel instability fully develops until saturation, with the rapid growth of the magnetic field, the thermal-ion Larmor radius become much smaller as $r_{gthi} \sim 40 \mu\text{m}$ and the drift-ion Larmor radius as $r_{gdrift} \sim 500 \mu\text{m}$ comparable with the interpenetrating scale. This latter condition is known as the Alfvén current limit. Here drift-ion's Larmor motions will significantly influent the velocity anisotropy. As a result, the Weibel instability cannot further develop and reach the stagnation as the reviewer's statement.

We have added discussions in our revised manuscript to clarify this important point [see item 4 in the "List of Changes"].

" In linear phase, the magnetic fields grow exponentially (as depicted in Fig. S3(d)) with initial magnetic perturbations $B_0 \sim 10^5 \text{ G}$. The thermal ion Larmor radius, $r_{gthi} = mu_{thi}/q\delta B_0 \sim 250 \mu\text{m}$ is much smaller than the transverse size ($L_y = 2 \text{ mm}$), while drift ion Larmor radius, $r_{gdrift} = mV_{flow}/qB_0 \sim 3100 \mu\text{m}$ is much larger than the interpenetrating depth ($L_x \sim 900 \mu\text{m}$). This leads to the motion of ions being characterized as Larmor precession, maintaining the velocity anisotropy to facilitate the development of WI. The ions in transverse directions (y) will be further converged by the magnetic perturbations resulting in the generation of stronger currents. After that, WI steps into the nonlinear phase, where the magnetic fields continue to grow through filamentary current coalescence process. In this phase, the drift-ion's Larmor motions become smaller as $r_{gdrift} (\sim 500 \mu\text{m})$ comparable with L_x when the magnetic field strength reaches 0.6 MG. This change will significantly disrupt the velocity anisotropy, leading to the saturation of Weibel magnetic fields."

(5) "Do the 2-D PIC simulations include Coulomb collisions?"

As stated in our manuscript, the mean free paths for ion-ion ($i-i$) and ion-electron ($i-e$) in interpenetrating flows significantly exceed the system size, hence Coulomb

collisions are not considered in our PIC simulations. As previous works [Nat. Phys. 11,173 (2015); Nat. Phys. 16, 916 (2020)], where the mean free path for $i-i$ and $i-e$ was also much greater than the system scale, they also did not consider Coulomb collisions. We have added an explanation of Coulomb collisions for PIC simulations in our revised manuscript [see item 5 in the “List of Changes”].

"And similar to previous works [Nature Physics, 2015,11(2):173-176, Nature physics, 2020, 16(9): 916-920], since the estimated average free path between particle collisions was much greater than the system scale, hence Coulomb collisions are not considered in our PIC simulations.

"

(6) *"The manuscript states: "At times later than that observed in our experiments (6ns). . .", implying that the experiment didn't last longer than 6 ns. However, Fig. 4c shows the P_1 and P_2 populations up to 16 ns. I presume that this is simulation data, but it's not clear from the text."*

Thank you for your useful comments. We are very sorry for this misleading description in our manuscript. In fact, the experimental time is also much longer than 6 ns. Firstly, as shown in the newly added Fig. R1 obtained by the streaked optical diagnostics, we have measured the evolution of the single flow free expansion (0-18 ns) and interpenetrating flow interactions (0-8 ns). Obviously, these plasma flows are sufficient to drive the evolution of the Weibel instability and the formation of the turbulent plasmas.

Fig. R1 Experimental measurement of the freely expanding flow and interpenetrating flows using time-resolved streaked optical diagnostics.

Secondly, as shown in the density distribution of 3D-RMHD simulation for a single flow [Fig. R2], from 15 ns to 25 ns, the plasma flow has not disappeared even at 25 ns. We also plot the density and velocity temporal profiles of a single flow at the midplane region obtained from the RMHD simulation [Fig. R3], we can observe that despite a decrease in velocity, the plasma flow remains present throughout. In the early stage, predominantly low-density high-speed plasma flow is observed, whereas in the later stage, it transitions to low-speed high-density plasma. Yes, the Figure 4c's data is from the PIC simulation. We are sorry for the misleading description in our manuscript, the plasma flow remains after 6ns, and we have modified the related texts in our manuscript.

To clarify this important point, we have added a short paragraph in the revised manuscript that reads [see item 6 in the "List of Changes"]:

"The measurement and corresponding simulation show that both interpenetrating flows last much longer than 15 ns, supporting our study of turbulence formation (see Supplementary Fig. 3 and Fig. 4)."

Furthermore, these important results for density and velocity temporal profiles as supplemental materials have been added in Supplemental Information.

Fig.R2 Density distribution of RMHD simulation results at late stage. Panels (a) to (c) show the density distribution of 3D-RMHD simulation for a single flow at t = 15, 20, 25 ns respectively.

Fig.R3 RMHD simulation results. It depicts the temporal profiles of density (solid line) and velocity (dashed line) for a singular flow at the midplane region. These profiles are derived from simulations of laser ablation conducted under the laser and target conditions identical to those of our experimental setup.

(6-1) "Also, Fig. 4c shows that the P_1 population only becomes suprathermal well after t = 6 ns (Stage 3 @ ~14 ns). In fact, the P_1 population isn't even suprathermal at the time the experiment ended (~6 ns)."

Thanks for the reviewer's comment. As we reply in Comment 6 of Reviewer #1, the experiment does not end at 6 ns. We are sorry again for this misleading information in our manuscript. We agree with the reviewer's comment, the P_1 only becomes suprathermal well after t=6ns (stage 3). We have added a short paragraph in the revised manuscript that reads: [see item 7 in the "List of Changes"].

"In stage 3, rapid acceleration occurs as the particle undergoes multiple collisions with magnetic islands, leading to a substantial energy boost from 20 keV to 80 keV, thereby transforming it into a suprathermal particle."

(7) "Given 6), if the acceleration doesn't even occur until much later than the experiment, how can it explain the purported power law in the observed electron energy spectrum?"

Thank you for your valuable feedback. Firstly, as noted in Comment 6 of Reviewer #1, it is important to note that the experiment duration extends beyond 6 ns. Secondly, the electron energy spectrum is obtained by electron magnetic spectrometer [Rev. Sci. Instruments 88, 053507 (2017)], which is time-integrated and use image plate as the detector. It allows to receive the escaped energized electrons throughout the entire experimental evolution period. As a result, our experiments enable us to effectively diagnose the generation of superthermal electrons and the power law electron energy spectrum.

(7-1) "Fig. 4b also suggests an issue, since the PIC-derived power law only really develops around $t \sim 20$ ns, which is consistent with the bulk of the acceleration occurring in Stage 3. The lines around $t = 6$ ns don't really show a power law in the required window of energies."

Firstly, we apologize for the misleading information in our manuscript regarding the experiment duration. As highlighted in Comment 6 of Reviewer #1, the experiment extends beyond $t = 6$ ns. We also concur with the reviewer's observation that the power law emerges around $t \sim 20$ ns. Furthermore, these spectra align with the particle trajectories in our Particle-in-Cell (PIC) simulation. Additionally, we have included a more detailed description of the temporal evolution of the energy spectrum in our revised manuscript as: [see item 8 in the "List of Changes"]

" Through PIC simulations, we have found that kinetic turbulent plasmas have the capability to accelerate electrons from the thermal pool to energies nearing 600 keV within the experimental timescale, as depicted in Figure 4b. Initially, the electron energy spectrum primarily displays a thermal pattern in the early stages, according to the PIC simulation results. However, as the turbulent plasma evolves, particularly in the later stages, approximately at $t \sim 20$ ns, the electron energy spectrum gradually transitions to a power law structure. Over time, the non-thermal electron distribution in the plasma region subjected to wave-particle interactions evolves towards a power-law energy tail, characterized by a spectral index of $p = -3.3$."

(7-2) "In fact, a plausible explanation for the two stream experimental spectrum in Fig. 4a, and the $t \sim 6$ ns PIC spectrum, is that they represent hotter, perturbed, Maxwellian distributions. This would make sense, as the merging of the streams would heat them (particularly if collisions aren't that negligible), and kinetic effects would then account for a deviation from a full Maxwellian distribution."

Thank you for your valuable comment. Firstly, as emphasized in our manuscript, the mean free paths for ion-ion and ion-electron collisions considerably exceed the system size. As noted by the reviewer, deviations from a full Maxwellian distribution due to heating through collisions typically arise when collisions are non-negligible. Consequently, both in our experimental setup and in the PIC simulations, the likelihood of observing a power-law spectrum attributable to particle collision effects may be relatively low.

Secondly, the PIC spectrum at $t \sim 6$ ns (Fig. 4b) might represent hotter, perturbed Maxwellian distributions. In the early stages of interaction, due to the strong velocity anisotropic distribution of plasma in our experiment and PIC simulations, the Weibel instability can be triggered, leading to the generation of strong magnetic fields. Plasma particles can interact with these strong magnetic fields, as demonstrated in Fig. 4c,

showing the energy change of P_1 particle in stages 1 and 2, which can contribute to the modification of the electron energy spectrum. However, for the two-stream experimental spectrum depicted in Fig. 4a, it represents the distribution of the number of electrons detected by the electron magnetic spectrometer throughout the entire interaction period (not only within 6 ns), which is a time-integrated one. Moreover, in the medium to high-energy range, the electron energy spectrum already exhibits a very pronounced power-law structure. Therefore, the generation of superthermal and non-thermal electron energy spectrum is predominantly due to interactions between electrons and turbulent magnetic fields.

We have incorporated a more comprehensive description of the temporal evolution of the energy spectrum in our revised manuscript [see item 8 in the "List of Changes"].

(8) "The self-similarity argument for translating the RMHD simulations to manageable, scaled-PIC simulations makes sense only if the underlying ideal MHD symmetry remains unbroken in the system. The authors note this by mentioning that viscosity, heat conduction, etc. break this symmetry, but that these transport effects shouldn't appear because the system wide Reynolds number, etc. are large. However, the authors experimentally infer that the magnetic correlation length becomes smaller than the system size, implying that the fluid closure scheme underlying ideal MHD is in question for the system at the relevant scales. For this reason, we may not expect a one-to-one correspondence between laboratory-scale RMHD simulations of the problem, and kinetic simulations at a reduced scale."

Thank you for your insightful comment. Firstly, we apologize for the misleading description provided. In our manuscript, the self-similar transformation involves converting the plasma parameters from radiation-magnetohydrodynamics (RMHD) to those required for particle-in-cell (PIC) simulations before the interaction of plasma streams. Essentially, the plasma parameters derived from RMHD simulations serve as the initial conditions for the plasma flow injected into the PIC simulation. Due to the high Reynolds number, magnetic Reynolds number, and other relevant dimensionless parameters in our setup, the ideal MHD symmetry remains unbroken prior to the interaction of plasma streams. As noted by the reviewer, the self-similar transformation between RMHD and PIC make sense only if the underlying ideal MHD symmetry remains intact in the system.

Specifically, prior to the onset of significant kinetic effects, RMHD is employed to model the macroscopic states of the plasmas. Subsequently, PIC is utilized to simulate the ensuing kinetic processes, with the injection of plasma velocity and number density based on the results obtained from the RMHD simulation. We concur with the reviewer that laboratory-scale RMHD simulations are unable to capture kinetic effects during plasma flow penetration.

We have revised the relevant description in our manuscript as: [see item 9 in the "List of Changes"].

"Initially, RMHD is employed to model the macroscopic states of plasmas, preceding the consideration of kinetic effects. Subsequently, PIC is utilized to simulate the ensuing kinetic processes. The injection of plasma velocity and number density in the subsequent stage is determined based on the results obtained from the RMHD simulation."

(9) "In summary, the results showing the structure of the magnetic turbulence -- which is believably the result of the ion Weibel instability (given the observed current filament evolution, and the correspondence between the theoretical Weibel growth rate and the experimentally inferred one) -- are compelling."

"However, the core result of the paper is built upon a shaky foundation. I am not convinced that the semblance of a power law in the observed electron energy spectrum points to SA as the only plausible explanation."

We thank the reviewer for recognition of the compelling nature of our results, which demonstrate the structure of magnetic turbulence arising from the ion Weibel instability. We also appreciated the reviewer's critical feedback on the interpretation of the observed electron energy spectrum. While we understand your skepticism regarding the interpretation of the observed electron energy spectrum as indicative of stochastic acceleration, we would like to provide additional context and clarification. Our assertion regarding stochastic acceleration as the explanation is based on a comprehensive analysis of the experimental data, acceleration theory and PIC simulations.

Firstly, it's important to note that the semblance of a power law in the electron energy spectrum is indeed a complex phenomenon and can potentially arise from various physical processes. Our interpretation of stochastic acceleration as the primary mechanism is supported by several factors:

1. Experimental data: To study the electron acceleration mechanism, three electron magnetic spectrometers placed at different observation angles are used to measure the angle distribution of the accelerated electrons, as shown in Fig. R4. One can find that the obtained electron spectrums from three observation angles show consistent power-law distributions, indicating that the accelerated electrons are almost isotropic. All these important data supporting our findings have been included in the revised manuscript and supplementary information.

Fig.R4 the angular distribution of non-thermal electron energy spectrum obtained from three different EMSs.

2. Acceleration mechanisms: It's true that many potential processes could contribute to the observed power-law distributions, such as magnetic reconnection, shock acceleration (diffusive shock acceleration and shock drift acceleration), turbulent acceleration (or stochastic acceleration), wave-particle interactions. (i) Magnetic reconnection, the accelerated electrons from MR are mainly in the parallel electric field and perpendicular reconnection electric field directions [ApJ 882, 143 (2019)]. (ii) Diffusive shock acceleration, the accelerated electrons from DSA are mainly in the shock propagation direction [Nat. Phys. 16, 916 (2020)]. (iii) Stochastic acceleration, the accelerated electrons from SA are almost random, depend on the turbulent plasmas [Space Sci Rev. 173, 535 (2012)]. From our observed accelerated electrons in three observation angles, it is convinced that the stochastic acceleration associated with turbulence is the promising mechanism contribution to the non-thermal electron generation.

3. PIC simulations: The corresponding simulations using the experimental parameters have well-reproduced our observed features including the turbulence and electrons with power-law distributions. After recording their motion trajectories in the whole evolution as shown in Fig. R5 and Fig. 4, one can see that thermal electrons collisions with quasi-magnetic-island structures are frequently. Each collision may gain energy from the magnetic fields, or may lose energy giving back the magnetic fields. A net energy gain will occur after frequent collisions, leading to the acceleration electrons to higher energy and formation of a power-law distributions. Two typical electrons are shown in Fig.4(c), the P_1 particle finally gain energy, while P_2 still thermal. And more particles are given in Supplementary, and as shown in Fig. R5.

Fig.R5 The typical particle trajectories in Weibel turbulent plasmas. The top panels (a and f) illustrate the kinetic energy histories of two selected electrons over time. The left panels (b to e) display the trajectories of particle 1 at different time intervals, while the right panels (g to j) depict the trajectories of particle 2. In each panel, the black stars indicate the initial positions of the particles, while the red stars indicate their

positions at the end of each interval.

4. Analysis and relevant works: As discussions in our manuscript and Reviewer #2's comments: "The reported experiments here were performed at a smaller laser facility (Shenguang II) using current experimental setup, and the plasmas did not have enough sizes (in the unit of ion gyro-radius or ion skin depth) to form a complete shock structure, but similar electron acceleration was detected (Fig.4a, as compared to Fig.4a of Ref.20). Therefore, the observed electron acceleration should be only by the plasma turbulence mediated by Weibel instability but without shocks."

Our findings are in line with previous research in similar conditions. As shown in the most recent works [J. High Energy Astrophys. 40, 1 (2023); ApJL 959, L8 (2023)], they found that the energy gain of the particles takes place when they bounce back and forth between converging turbulent magnetic fields. The particles can be efficiently accelerated in self-driven turbulent reconnection. As shown in Fig. R6, their results show that the test particle is trapped in a small region before $\Omega t \sim 200$, which corresponds to the mirror reflection at the mirror points. After $\Omega t \sim 200$, due to the random changes in the magnetic field (turbulent magnetic field), we can clearly see the transition from the slow mirror diffusion within a small region to the fast-scattering diffusion over a large distance, leading to rapid changes in particle energy.

Fig.R6 Gyroaveraged trajectory of a test particle. The image excerpted from [Zhang, Chao, and Siyao Xu. The Astrophysical Journal Letters 959.1 (2023): L8] published under a CC BY 4.0 license <https://creativecommons.org/licenses/by/4.0/>.

This result is similar to our PIC simulation results. In our simulation results, particles accelerated to high energies are initially confined to a small region by strong magnetic fields, where they undergo cyclotron motion. However, as the particles move into regions where the magnetic field structure changes, they become detached from the strong magnetic confinement and interact with other turbulent magnetic fields, resulting in significant changes in energy.

In summary, it is reasonable to believe that stochastic acceleration is one promising mechanism contributing to these energized electron generation in our experiments. To clarify this important point and make more our work convince, a

significant amount of additional content and experimental data have included in the revised manuscript and Supplemental Information [see item 10 in the "List of Changes"].

(10) "This work should be published, but it may not be appropriate for Nature Communications. While the observed magnetic and electron spectra offer a valuable glimpse into the dynamics of Weibel turbulence, the results do not go far beyond previous works (e.g., Mondal et al.). If a more convincing case could be made for the SA mechanism, I may change my mind."

Thank you for your thoughtful evaluation of our manuscript. We appreciate your acknowledgment of the value of our work in providing insight into the dynamics of Weibel turbulence. In the above response, we addressed all your comments point-by-point. We hope that our response has addressed your concerns.

The previous work (Mondal et al, 2012) primarily focuses on the process of strong magnetic field formation under relativistic laser intensities ($I > 10^{18}$ W/cm²). The formation of the filamentary magnetic field structures mainly due to the filamentation of the laser produced electron beams and the associated cold return currents develops due to the Weibel-like instability. In our study, as we utilize ns-lasers, we pay attention to the dynamics of ions. In the experimental setup, the formation of turbulent magnetic fields predominantly relies on the development of ion Weibel instability. Therefore, compared to previous studies, there are differences exist in both the formation mechanisms and kinetic dynamic characteristics of the filamentary turbulent magnetic fields and electron acceleration processes observed in our experiments.

Regarding the stochastic acceleration (SA) mechanism, we agree that strengthening the evidence and compelling nature of this aspect could potentially improve the overall significance and impact of the work. In the revised manuscript, we will aim to provide a more thorough and rigorous analysis of the SA process. As shown in comment 9 of Reviewer #1, we obtained electron energy spectra using electron spectrometers oriented in various directions. The experimental results indicate that the directions of accelerated high-energy particles are random, lacking specificity, which closely aligns with the characteristics of stochastic acceleration. Furthermore, a comparative analysis of our PIC simulation results with those of previous studies demonstrates that turbulent acceleration is the primary source of super-hot electrons in both our experiments and simulations. We hope that these modifications will meet the requirements of the reviewer [see item 10 in the "List of Changes"].

II. Responses to Reviewer #2:

(1) "This manuscript reports new results from laboratory experiments on electron acceleration by kinetic plasma turbulence generated by Weibel instability. Particle acceleration has been an outstanding problem for plasma astrophysics for a long time and most of the relevant research has been performed either theoretically, numerically, or observationally. The reported laboratory results provide a much-needed test of some of the proposed acceleration mechanisms. Previous experiments were performed on a larger laser facility (NIF) which generated Weibel instability-mediated collisionless shocks, which led electron acceleration (Ref. 20). The reported experiments here were performed at a smaller laser facility (Shenguang II), and the plasmas did not have enough sizes (in the unit of ion gyro-radius or ion skin depth) to form a complete shock structure, but similar electron acceleration was detected (Fig.4a, as compared to Fig.4a of Ref.20). Therefore, the observed electron acceleration should be only by the plasma turbulence mediated by Weibel instability but without shocks. The measurements of this plasma turbulence shown in Fig.2 and Fig.1a are of high quality, which have been analyzed for their spectra shown in Fig.3k. These results are sufficiently important and should be published in Nature Communications."

Thank you for the thoughtful and detailed review of our manuscript. We are very pleased to receive such a positive assessment of the significance and quality of our experimental results on electron acceleration in turbulence arising from Weibel instability. Your feedback underscores the importance of our work in providing much-needed laboratory validation of the plasma astrophysics theories and computational models describing particle acceleration in turbulent environments.

(2) "However, I would like to offer the following suggestions for authors to improve their manuscript."

(2-1) "The author should highlight the fact that similar electron acceleration was observed without collisionless shocks. It is difficult for readers to find out from the current version of the manuscript about this important difference unless they read Ref.20 and compared. Please note that this does not reduce the importance of the results reported in Ref.20 nor by this manuscript, and there is an important physics insight (that similar electron acceleration occurs with and without shocks) by comparing both experiments."

Thank you for this valuable suggestion. You make a very good point that we should more explicitly highlight the key distinction between the electron acceleration observed in our experiments compared to the previous work on collisionless shocks. To highlight the results that electron accelerated by the turbulent magnetic island instead of collisionless shock, we have added the following text in the revised manuscript [see item 11 in the "List of Changes"].

“Recent experiment performed at National Ignition Facility (NIF) have showed that

non-thermal electrons with power-law distribution are produced by turbulent shock via 1st order Fermi acceleration. However, in our experiments the obtained Weibel correlation length ($\sim 9c/\omega_{pi}$) is significantly lower than the theoretical prediction distance of supporting Weibel-mediated shock formation ($\sim 100c/\omega_{pi}$). Therefore, the shock acceleration mechanism can be ruled out in our findings.”

(2-2). *It is important to have control cases with only single flow as shown in Fig.4a and in Fig.1c. Please indicate which single flow between a flow from the negative x direction and a flow from the positive x direction. Please also show results from the single flow from another direction of X-axis. Since the ESM has an angle of 70 degrees from the X-axis, these two measurements are unlikely the same so showing both single flow cases are important.*

Thanks for your valuable comments. As the reviewer said, because the ESM has an observation angle of 70 degrees relative to the flow propagation direction ($\pm x$), the measured thermal electron features are different. One is the thermal temperature, we can see that the measured temperature in opposite-side flow ($-x$) is higher than that in same-side flow ($+x$); the other one is the number of thermal electrons, the observation angle leads to that the electron number from the same-side flow ($-x$) is two orders of magnitude lower than that from the opposite-side flow ($+x$). These measured typical thermal electron spectrum from both sides have been added in Fig. 4a and Fig.1c. To highlight the different between thermal and accelerated electrons, we have added experimental data and corresponding descriptions in the Supplemental Information [see item 12 in the "List of Changes"].

“For comparison, two thermal electron backgrounds (< 0.3 MeV) with an approximate Maxwell distribution are obtained from two distinct single flows. One background is from the flow propagation in the $+x$ direction, the other one is from the flow propagation in the $-x$ direction.”

Fig.R7 thermal electrons spectrum from different single flows.

(2-3) "According to numerical simulations, plasma turbulence should be much stronger at $t=6\text{ns}$ (Fig.3), especially including the so-called "magnetic islands", and electron energetic tail takes a long time to develop, as long as 15-25ns (Fig.4b, Fig.4c), but experimental data are shown only until $t=3\text{ns}$. Are there experimental data supporting this important development?"

As we reply to the Reviewer #1's comment 6, the lifetime of single plasma flow and interpenetrating flows are as long as the simulation time ($\sim 20\text{ ns}$), shown in Fig. R1. The corresponding simulation is well reproduced the flow conditions in the whole experimental time.

We also tried to measure the later evolution of the turbulent plasma, especially the "magnetic island" using the streaked optical diagnostics. However, limited by the optical diagnostics due to the "3D effect", the probe of optical diagnostics passing the plasma density is much lower than the critical density. Normally, it is about $0.01N_c \sim 5 \times 10^{19}\text{ cm}^{-3}$, where N_c is the critical density dependent on the probe wavelength of 527 nm. Supported simulations show that the plasma density is about 10^{20} cm^{-3} when the magnetic island formation in the much later turbulent plasmas. Consequently, it is hard to directly observe the developed magnetic island features.

(2-4) " V_{flow} is used in line 137 while u is used in line 141. Please check."

We have unified the expression of flow velocity using V_{flow} in the revised manuscript.

(2-5) *"Please cite some references on Weibel instability-mediated turbulence – experimentally and numerically."*

Thanks for your suggestions. As we clarified in our response to the Reviewer #1's comment 3, we have cited some representative references on Weibel-mediated-turbulence including experiment and simulation, such as,

"S. Mondal et al., Proc. Natl. Acad. Sci. 109, 8011 (2012)"

"C. M. Huntington et al., Nat. Phys. 11(2), 173–176 (2015)"

"G. Chatterjee et al., Nat. Commun. 8, 19750 (2017)"

"M. Zhou et al., ApJ 960, 12 (2023)"

"C. K. Li et al., PRL (2019)"

III. Responses to Reviewer #3:

(1) "The manuscript "Electron stochastic acceleration in laboratory-produced kinetic turbulent plasmas" presents results of laboratory experiments on laser-produced plasmas that demonstrate electron acceleration associated with development of Weibel instability in interpenetrating flows. Based mainly on results of supporting computer simulations, it is argued that acceleration is stochastic and is associated with reflection by magnetic structures."

"Overall, I believe that the manuscript contains results that merit publication in Nature Communications in the sense that they are sufficiently novel and are of interest to a broad community of researchers. However, I also feel that major improvements are needed before the manuscript becomes suitable for publication."

Thank you for your thorough and constructive review of our manuscript. We appreciate your overall assessment that the results merit publication in Nature Communications, while also providing valuable feedback on where significant improvements are needed. We agree that the manuscript requires major revisions before it will be suitable for publication in Nature Communications. Your detailed comments and suggestions are extremely helpful for improving our manuscript. Subsequently, we have provided point-to-point responses to your comments to address them comprehensively.

(2) "The manuscript, especially abstract and the introduction, could benefit from careful proof-reading and editing. Many of the sentences are poorly constructed. I point out a few of those below, but it's not the job of a reviewer to proof-read and edit the text."

Thank you very much for your kindly review. We tried our best to improve the manuscript and made some changes to the manuscript. These changes will not influence the content and framework of the paper. And we marked them in red in the revised manuscript.

(3) "Out of the 4 Figures in the main section, I believe 3 need improvements (summarized below)"

In the revised manuscript, we have modified the figures according to the reviewer's advices.

(4) "The manuscript places a great deal of attention on the stochastic nature of the observed electron acceleration. While this might be true, the relevant presentation focuses on analysis of 2 electron trajectories in the simulation. Some statistical evidence is required to demonstrate that stochastic acceleration by turbulence is indeed the dominant mechanism."

We appreciate the reviewer's insightful comment. Perhaps, our expressions lead

to a misunderstanding. Here we want to express that two kinds of typical electrons have been selected to verify the stochastic acceleration. In the revised manuscript, we clarify this mechanism as a promising acceleration basing on a comprehensive analysis of the experimental data coupled with theoretical considerations. As we reply to the Reviewer #1's comment 9, we have provided the electron spectra in three different observation angles. Three independent spectra show a similar power-law index and the maximum almost reaching the threshold of ESM. It indicates that the energized electrons demonstrate isotropy, which manifest the stochastic acceleration.

(5) "Line 62: "magnetic reconnection acceleration". Please be more specific. There has been a great deal of work (and some controversy) on the exact mechanisms accelerating particles in magnetic reconnection regions. Some examples include acceleration by parallel electric field, stochastic acceleration by contracting magnetic island, curvature drifts, etc"

"In the same sentence: the first part reads "astronomical observations...indicate...", while the second part says "...it would be difficult to identify from observations". This needs some re-phrasing. Perhaps one can say that observations indicate acceleration takes place, while theory and simulations identified various mechanism mentioned in this sentence."

Thank you very much. In the revised manuscript, we have clarified the reconnection acceleration mechanism and re-phrase the abstract as: [see item 13 in the "List of Changes"].

"Astronomical observations have provided insights into particle acceleration in various environments, but identifying which specific mechanism operates is difficult. Numerical simulations have proposed that magnetic reconnection, shock acceleration, and stochastic acceleration play roles in generating high-energy particles. Recent experiments have confirmed that electrons can be accelerated to relativistic speeds by parallel electric field, perpendicular reconnection electric field and collisionless shock"

(6) "Line 70 and throughout the manuscript: in what sense is "self-organized Weibel instability" being used? It might be appropriate to say "turbulence originating from Weibel instability" or something along those lines."

Thank you very much. we have modified these expressions with turbulence originating from Weibel instability throughout the manuscript.

(7) "Line 75: I fail to see how these results imply that stochastic acceleration may be an efficient mechanism in various environments. That would depend on the type of turbulence being produced, etc; Here and in many other places throughout the text I get the impression that the rigor of presentation is being sacrificed to make the results more broadly appealing. I think this is a mistake."

Thank you very much for these kind suggestions. We have modified these descriptions as: [see item 14 in the "List of Changes"]

" This study sheds light on a critical transition period during supernova explosion, where kinetic turbulences emerge prior to Weibel-mediated-shock formation. Our results also suggest that electrons undergo stochastic acceleration during this transition phase."

(8) "I also don't see how the manuscript addresses the issue of particle injection into other acceleration processes. That scenario has been proposed before and in my opinion the manuscript does not add anything specific apart from showing that Weibel instability can indeed accelerate particles as expected. The latter is a very good experimental result, but it does not add to our understanding of the pre-acceleration problem."

Thank you for your insightful comments. We appreciate you raising the important issue of particle injection problem into other acceleration processes. In our study, we primarily focused on experimentally demonstrating the Weibel instability associated with turbulence as an effective mechanism for electron acceleration, which have previously been proposed in theory but not conclusively observed in the laboratory. While we agree that our work does not directly address the injection problems, we believe that our findings contribute to a better understanding of the fundamental physics underlying particle stochastic acceleration in astrophysical environments.

(9) "Lines 102, 105: these sentences need editing (along with many others in the manuscript)"

Thank you very much. we have rephased these sentences as: [see item 15 in the "List of Changes"]

"Although several acceleration scenarios have been proposed, there still exist numerous uncertainties regarding individual mechanisms, for example, the onset of MR with multiscale, the injection problem of DSA, and so on."

"Motivated by these astrophysical challenges, laboratory experiments with scaled-down versions provide a novel approach to study them in details."

(10) "Line 111: "externally applied", or "external"? Also, I would not say "astrophysical magnetized shocks". At best, one can say that such shocks have parameters relevant to astrophysics, but even that is highly debatable (e.g. system sizes in all experiments are tiny even in normalized units)"

Thank you very much. we have taken the reviewer's advice and modified these expressions as: [see item 16 in the "List of Changes"]

"Shocks with parameters relevant to astrophysical ones are produced using laser-driven piston plasma expanding into ambient gas under an externally applied magnetic field"

(11) *"Lines 116 & 117: perhaps acceleration efficiency needs an explicit definition here."*

We have defined the acceleration efficiency $\Delta E/E$, the average fractional energy gain per collision between electrons and scatters.

(12) *"Line 131-132 or somewhere in the description of Fig 1: I think it would be good to bring here some information from the "Methods" section on what each individual diagnostic measures."*

We have added the important information about the description of Fig. 1 into the text [see item 17 in the "List of Changes"]

" The interferometer was used to measure the plasma density basing on the plasma refractive index of $N \approx 1 - n_e/n_c$, where n_e is the plasma density and n_c is the critical density of the probe. The shadowgraphy was used to measure the denser filaments. A modified Faraday rotation method was used to measure the topology and strength of the Weibel magnetic fields."

(13) *"Table 1: it might be good to also include electron mean free path or collisional frequency."*

As the Reviewer #1 also give this suggestion, we have added these important values in the revised manuscript. One can see that the mfp between e-e is much smaller than the target size, indicating that the high collision frequency between e-e can be regarded as a thermal electron background. All these important parameters has been summarized in the revised manuscript and supplementary information.

(14) *"Line 143: I am assuming that the growth rate is a theoretical value: if yes, please specify that. In the method section, the description of how the growth rate was obtained is limited to simply stating that it comes from an electromagnetic solver. Please describe the solver being used (what equations are being solved?), parameters used in the calculation, and the equilibrium assumed."*

Yes, in our experiments, the linear growth rate of Weibel instability is derived from the dispersion relation of plasma instabilities for unmagnetized interpenetrating flows ($\pm x$), which has been extensively explored in many previous papers [Phys. Plasmas 17, 032114 (2010); Phys. Rev. Lett. 132, 155103 (2024)]. Here we focus on the ion-driven Weibel instability, which grows perpendicularly to the flow propagation direction. In this scenario, the electrons have established a thermalized background with the conditions of $V_{the} \gg V_{flow} \gg V_{thi}$. The simplified dispersion relation for Weibel instability can be expressed as follows: [Nat. Phys 11, 173 (2015)],

$$k_y^2 c^2 + \omega_{pi}^2 / (1 + |k_y| / \Gamma \sqrt{2T_e / \pi m_e}) + \omega_{pi}^2 [G_1(\Gamma^2 A m_p / (2k_y^2 T_i)) - k_y^2 V_{flow}^2 / \Gamma^2 G_2(\Gamma^2 A m_p / (2k_y^2 T_i))] = 0.$$

Here c represents the speed of light, k_y is the most unstable wave number perpendicular to the flow direction, V_{flow} is the flow velocity, A is the atomic number, m_p is the proton mass, Γ is the linear growth rate. Additionally, G_1 and G_2 are the dimensionless functions defined for $x > 0$ as

$$G_1(y) = 1/\sqrt{\pi} \int_0^\infty y e^{-x^2} / (x^2 + y) dx$$

$$G_2(y) = 2y/\sqrt{\pi} \int_0^\infty x^2 e^{-x^2} / (x^2 + y) dx$$

Taking experimental parameters into above dispersion equation, we obtain the relationship between the linear growth rate Γ and the wave number k_y , as shown in the Supplementary Figure 3. The most unstable mode of Weibel instability observed in our experiments is $k_y \sim 1/(L_{spacing}) = \omega_{pi}/c$, that is to say $k_y c / \omega_{pi} \sim 1$. Based on data presented in the Supplementary Fig.3(a) and 3(b), we can obtain the linear growth rate is $\Gamma \sim 0.1 V_{flow} / c \omega_{pi} \sim 1.5 \times 10^9 \text{ s}^{-1}$.

In the revised supplementary information, we have added the necessary information about dispersion equation, typical parameters, and functions [see item 18 in the "List of Changes"].

(15) "Fig 1:

- a. Color bars in panel a) are unreadable, unless one zooms in. It is possible to that on screen, but it would not be possible in print.
- b. The vertical axis in panel c) needs a label.
- c. Line 153 in the caption: "No signal appearing in the central region indicates...": - this needs an explanation.
- d. I would also suggest labeling the left-most schematic (i.e. labels would go a-d instead of having one un-labeled sketch)"

Thank you very much. According to the reviewer's suggestions **a**, **b**, and **d**, we have improved the new Fig. 1 as follows,

According to the reviewer's suggestions **c**, we have modified the corresponding caption as: [see item 19 in the "List of Changes"]

"The detected X-ray self-emission are mainly from the bremsstrahlung caused by Coulomb collisions. The X-ray signal disappears in the interaction region, indicating that the Coulomb collisions between interpenetrating flows are weak and can be ignored."

(16) "Lines 159 -160 and in other places "the space ... almost keeps constant" – is it better to say "the spacing ... is almost constant"?"

Thank you very much. We have modified this expression as "The spacing between filaments is almost constant (comparable with the ion inertial length $\sim c/\omega_{pi}$)".

(17) "Lines 161-163: it's not exactly clear how the linear stage is identified. Is the time-resolution of this diagnostic 0.5 ns? Or are these two snapshots chosen as the most representative? If the former, the presented growth rate is a lower bound at best, since the growth time can be much shorter. If the second, perhaps one can show evolution of an average magnetic field magnitude?"

Thank you for your valuable feedback. In our experiments, we choose two snapshots as the most representative cases. In the experiment, we collectively observed data at five time points: 0.5, 1.0, 2.0, 3.0, and 4.0 ns (due to issues with the probe light, the data quality at the 4.0 ns time point was compromised). The evolution of average Weibel magnetic field strength is obtained from experiments and simulations, as shown in Fig. R8. In the linear stage (0~2 ns), the growth follows

exponential form of $B \sim B_0 e^{\Gamma t}$, where Γ is the linear growth rate. After stepping into nonlinear phase, the magnetic fields quickly saturate. This important data has been added in the supplementary Fig. 3(d).

Fig.R8 the evolution of average Weibel magnetic field strength obtained from experiments and simulations

(18) "Fig. 2: please label axes. I would also add in the caption some information on what panels a) to d) represent (density? current?)"

Thank you for your useful suggestions. We have added the axes in Fig.2, also as shown in Fig. R9 as follows. Filaments in **a-d** measured using shadowgraphy represent the information of plasma density and in **e-f** measured using Faraday method represent the path-integrated magnetic field strength. We have added the information for panels (a) to (f) in the caption [see item 20 in the "List of Changes"].

Fig.R9 The modified Fig. 2 in our revised manuscript. Panels (a) to (d) represent the information of plasma density, and panels (e) and (f) show the distribution of magnetic fields at time 1.0 ns and 3.0 ns.

(19) *"Lines 179-180: "scaled-down two dimensional ..." is probably better than "scaling-down two dimensional"*

Thank you for your suggestion, we have modified it in our revised manuscript.

(20) *"Line 190: since the subsequent discussion appeals to magnetic islands, it might be useful to show the magnetic field topology (e.g., iso-contours of vector potential in 2D or something equivalent)."*

Thank you for your valuable feedback. We concur with the reviewer's observation regarding the utility of iso-contours of the magnetic vector potential (A) in 2D to elucidate magnetic field topology. Accordingly, we have incorporated them into Figure 3 in the revised manuscript. The vector potential A is intricately linked to the current density J through the equation $\nabla^2 A = -\mu_0 J$. Subsequently, the relationship between

current density J and magnetic field B is represented by $J = \frac{1}{\mu_0} \nabla \times B$. Consequently,

we can infer the magnetic vector potential distribution based on the magnetic field distribution and the equation $\nabla^2 A = -\nabla \times B$. Given the limitations inherent in our two-dimensional simulation, we are only able to obtain the A_x and A_y components.

Consequently, the total magnetic vector potential is obtained as $A_{tot} = \sqrt{x_2^A + y_2^A}$. The

distributions of total magnetic vector potential are illustrated in Fig. R10. In Fig. R10(a) and R10(c), the black, red, and gold lines delineate the iso-contours of the magnetic vector potential, signifying values of 2×10^{10} , 3×10^{10} , $4 \times 10^{10} \text{ Gs} \cdot \text{cm}^{-1}$, respectively. Notably, we observe the emergence of distinctive magnetic island-like structures around 3 ns. Furthermore, given the two-dimensional nature of our simulation, the presence of magnetic flux-tube structures within the xz plane is plausible. Subsequently, at 6 ns, the disturbed region's overall scale expands, concomitant with the appearance of additional magnetic island-like structures, indicative of heightened plasma turbulence.

Fig.R10 The iso-contours of the magnetic vector potential. Panels (a) and (b) display the distribution of magnetic fields and magnetic vector potential at 3.0 ns, respectively, while panels (c) and (d) exhibit the corresponding simulation outcomes at 6.0 ns. In panels (a) and (c), the black, red, and gold lines denote the iso-contours of the magnetic vector potential, indicating values of 2×10^{10} , 3×10^{10} , $4 \times 10^{10} \text{ Gs} \cdot \text{cm}^{-1}$ respectively.

We have added the iso-contours of the magnetic vector potential to Figure 3 and corresponding description in the revised manuscript, [see item 21 in the "List of Changes"].

"The magnetic vector potentials (A) in Fig. 3(f) and Fig. 3(j) are plotted by the iso-contour lines, which is intricately linked to the current density J through the equation $\nabla^2 A = -\mu_0 J$. We observe the emergence of distinctive magnetic island-like structures around 3 ns. Subsequently, at 6 ns, the disturbed region's overall scale expands, concomitant with the appearance of additional magnetic island-like structures,

indicative of heightened plasma turbulence. "

(21) *"Lines 219-223: I think the assumption that all kinetic-scale plasma turbulence is characterized by the same spectrum is not universally accepted, so appealing to it may be ill-advised. More broadly speaking, I would also say that the turbulence in these experiments is very different from the solar wind. Here, it is strongly compressible, originating from Weibel instability, and is presumably characterized by fluctuation magnitude δB comparable with the mean field B_0 . In the solar wind, the kinetic-scale turbulence is believed to be dominated by kinetic Alfvén fluctuations, is characterized by very weak fluctuation amplitudes $\delta B/B_0 \ll 1$ (at kinetic scales), is nearly perpendicular, etc."*

Thank you very much for the insightful suggestions. we have modified these expressions about the turbulence comparison between experiment and solar wind [see item 22 in the "List of Changes"].

" Although kinetic turbulence is also observed in solar wind, the mechanism is different from that in our experiments. The turbulence in our experiments, which originates from Weibel instability, is highly compressible, and presumably characterized by fluctuation magnitudes (δB) comparable with the average magnetic field (B). In contrast, the kinetic-scale turbulence in solar wind is dominated by kinetic Alfvén waves and characterized by $\delta B/B \ll 1$ "

(22) *"Discussion in lines 233-234: perhaps it is also good to mention (and describe) relevant simulation results here?"*

Here we discuss the ion motions to show that the ions are hard magnetized due the ion gyro-radius comparable with the turbulent region scale. Previous simulations also shown that Weibel-mediated-shock can be produced in turbulent region with needed width approximately $\sim 100c/\omega_{pi}$, much larger than that obtained ($\sim 9c/\omega_{pi}$) in our experiments. As a result, the collisionless shock cannot be produced and the shock acceleration mechanism can be ruled out in our case.

In order to clarify the non-thermal electron generation, we focus on the trajectories of accelerated electrons, as illustrated in Fig. 4(d)-4(f). One can find that electrons obtain a net gain after multiple collisions with magnetic islands. The overall process manifests a stochastic acceleration.

(23) *"Fig 4: I have lots of comments here:*

a. Panel a): one can argue that single flow spectra are also characterized by the same power-law in the range 200-300 eV, even if amplitude is lower.

- b. Panel b): why not use the same energy units as in panel a)?*
- c. Panel b): could you comment on why the time to achieve the power-law spectra is substantially longer in the simulation compared to the experiment?*
- d. Panels d)-f) are incomprehensible for two reasons: a) use of color (I am a colorblind person and cannot see the trajectories); b) they are too small; I believe it is common practice to try to make figures accessible."*

We sincerely appreciate the reviewers' insightful and constructive comments. For question (a), here we use the ESM to measure the thermal electron spectrum from the single flow. As we response to the Reviewer #2's comments, we have added a new thermal electron spectrum from the other flow. In this new figure, we have re-corrected the energy spectrum by considering the measurement error induced by the sensitivity of IP and the energy error from ESM. The total error in electron number is estimated as 15-20%. After this calibration process, one can see that the electron spectrum in the range of 200-300 eV from single flow is also near to the Maxwell distribution.

For question (b), we sincerely apologize for the oversight. The energy unit in panel (b) should have been the same as in panel (a); we forgot to adjust the energy unit. In our initial analysis, we normalized the units using the thermal energy of the shocked plasma ($k_b T_e \sim 2$ keV). However, subsequently, we converted the units to keV but forgot to adjust them in the figure. We deeply regret this mistake and have made the necessary adjustments in Figure 4.

For question (c), we apologize for the lack of clarity that led to your misunderstanding here. In fact, the data obtained by the electron spectrometer in the experiment is an integral quantity, which receives electrons throughout the entire experimental evolution period. Therefore, the exact time when the electron energy spectrum in the experimental data transitions to power-law spectra is indeterminate. However, we can confidently say that this transition doesn't occur significantly earlier than the time when the energy spectrum transitions to power-law spectra in the simulation, primarily since turbulent activity hasn't fully developed during the early stages of optical diagnostics.

For question (d), we sincerely apologize for the inconvenience caused by the inappropriate color choice that hindered your ability to see the particle trajectories. We have made the necessary adjustments, including boldening and enlarging the lines representing particle motion. The updated particle trajectory motion is depicted in Figure R11. We apologize once again for any inconvenience this may have caused.

We have modified the related figures and context in the revised manuscript [see item 23 in the "List of Changes"].

Fig. R11 The new images of the trajectory of particles during different acceleration stages.

(24) *"More broad comment about results summarized in Fig. 4: all said and done, this is analysis of two electrons that shows acceleration events that are characterized by large jumps in position. If one is to make a point that dominant acceleration mechanism is stochastic, some statistical analysis of many electrons is needed. I would probably suggest focusing this analysis on energetics of individual events."*

Thank you for your insightful comment regarding the results presented in Fig. 4. In Figure 4, we depict two types of particles. Particle P_1 serves as a representative of accelerated particles, whereas particle P_2 represents particles confined within a localized range and not subject to acceleration. In the analysis of Figures 4(g) and (h), we focus solely on the energy variation process of the accelerated particle P_1 . We strongly agree with the reviewer's perspective and suggest refining this analysis to focus on the energetics of individual events. In Figure 4, P_1 and P_2 are analyzed as contrasting particles for comparison. In our revised manuscript, we have revisited the explanation regarding the significance of particles P_1 and P_2 representing distinct types of particles [see item 24 in the "List of Changes"].

"As depicted in Fig. 4(c), P_1 represents the electrons ultimately accelerated to high energies, delineated into three acceleration stages according to their energy variations over time. Meanwhile, P_2 denotes thermal electrons exhibiting energy oscillations between 0 and 10 keV, and P_2 is also a typical representative of particles that have not been accelerated."

(25) *"Lines 286: "the average energy gain...": is this a result of an analysis of the simulation or an assumption? If the former, please describe the analysis."*

This result was obtained through analysis of our numerical simulation results. We sincerely apologize for not providing a detailed explanation of this in the manuscript. As shown in Fig. 4(h) in our manuscript, the average energy gains experienced by electrons in each reflection in Fig. 4(g) are plotted. We found that when the average velocity of particles reverses in the x-direction, there is a certain degree of linear fitting relationship between the energy change of particles and the particle energy at that moment. The proportionality factor is approximately 0.1, which is similar to the value of $V_A/c \approx 0.1$ (where V_A is the Alfvén speed obtained by the average magnetic field strength of 0.6 MG). We have added the analysis in our revised manuscript: [see item 25 in the "List of Changes"].

"As illustrated in Fig. 4(h), according to the simulation results, the average fractional energy gains experienced upon each reflection are $\Delta E/E \approx V_A/c \approx 0.1$ (where V_A is the Alfvén speed obtained by the average magnetic field strength of 0.6 MG). "

(26) *"The same comment about line 288: "the acceleration efficiency becomes first order in these multiple magnetic islands". Does this statement apply to this particular simulation, or is it a generic statement based on Ref. 22?"*

Thanks for your comment. The line 288 "the acceleration efficiency becomes first order in these multiple magnetic islands" is a generic statement based on Ref. 22. The particle acceleration efficiency values obtained in our simulation are similar to the results reported in Ref. 22.

(27) *"Line 292: "although both systems ..." does "both" here refer to laboratory and astrophysical plasmas? There is a wide variety of systems in laboratory and especially in astrophysics that are characterized by vastly different parameters, acceleration mechanisms, etc. I think this statement is too generic."*

We strongly agree with the reviewer's opinion. In our revised manuscript, we have modified this paragraph [see item 26 in the "List of Changes"].

" Similar dimensionless parameters between laboratory and supernova plasmas (as illustrated in Supplementary Table 2), allow our study to probe a transition phase of supernova explosion. This phase occurs when turbulence have generated before the formation of shock formation as the ejecta sweeping up the interstellar medium. Our experimental results provide a possibility that the process of electron stochastic acceleration associated with turbulence can take place in this transition period."

(28) *"Line 308: As I already stated earlier, I don't think it's a good idea to equate Weibel turbulence to solar wind turbulence. I don't believe Weibel turbulence teaches us anything about dissipation in the solar wind and warm interstellar medium, and likely in other systems listed there."*

We thank the reviewer's insightful suggestion and we total agree with the reviewer's view. In the revised manuscript we have removed these inappropriate descriptions.

(29) *"I think the description of the scaling assumption needs to be simplified. It is long and confusing. The kinetic equations are clearly not self-similar under the specified scaling. One can simply state, as is the common practice in the field, which dimensionless parameters are kept the same between simulations and real system and which ones are changed."*

We completely agree with the reviewer's opinion. Your suggestion to focus on specifying which dimensionless parameters remain constant between simulations and the real system, aligns well with common practices in the field. In our revised manuscript, we have significantly simplified the description regarding scaling assumption: [see item 27 in the "List of Changes"].

" This scaling ensures that plasma properties, including β , β_{ram} , the Mach number, and the Alfvén Mach number, the ratio of ion cyclotron times between the two systems ($\omega_{ci,1}^{-1}/\omega_{ci,0}^{-1}$) et al., remain conserved between RMHD and PIC. The transformation equations are,

$$r_1 = \sqrt{f_m/f_n}r_0, \rho_1 = f_m f_n \rho_0, p_1 = f_n f_T p_0$$

$$V_1 = \sqrt{f_T/f_m}V_0, B_1 = \sqrt{f_n/f_T}B_0, t_1 = f_m/\sqrt{f_n/f_T}t_0$$

where r , ρ , p , V , B and t respectively represent length, density, pressure, velocity, magnetic field and time, and more details of this transformation can refer to our previous work"

(30) *"Line 489 "the mean charge state remains consistent"; Does this imply that the charge state of individual ions can change in the simulation?"*

In transitioning from RMHD to PIC simulations, we maintain a consistent mean

charge state in the PIC simulations, while in the subsequent evolution, the charge state of individual ions varies based on the local conditions in the PIC simulations.

(31) *"Please provide complete information necessary to reproduce the simulation: parameters of the injected plasma (flow speed, temperature, density); how the simulation is initialized; how the injection is done. The goal should be ensuring that the simulation can be reproduced by an interested researcher. It is not possible based on the information provided. As I mentioned earlier, the same comment applies to the description of the linear growth rate calculation."*

We greatly appreciate the valuable suggestions provided by the reviewer. In our revised manuscript, we provided more details of our simulation. The simulation box dimensions after transformation are $L_x = 6$ mm and $L_y = 3$ mm, with plasma injection regions established on both sides. The density and velocity of the injected plasma are determined based on RMHD simulation results and experimental data, as illustrated in Fig. R3 in the Response letter. The simulation box is divided into grids of 2000×1000 , corresponding to a resolution of $0.03d_i$. We employ 100 macro-particles per species within each grid cell.

In our experiments, the linear growth rate of Weibel instability is derived from the dispersion relation of plasma instabilities for unmagnetized interpenetrating flows ($\pm x$), which has been extensively explored in many previous papers [Phys. Plasmas 17, 032114 (2010); Phys. Rev. Lett. 132, 155103 (2024)]. Here we focus on the ion-driven Weibel instability, which grows perpendicularly to the flow propagation direction. In this scenario, the electrons have established a thermalized background with the conditions of $V_{the} \gg V_{flow} \gg V_{thi}$. The simplified dispersion relation for Weibel instability can be expressed as follows: [Nat. Phys 11, 173 (2015)],

$$k_y^2 c^2 + \omega_{pi}^2 / (1 + |k_y| / \Gamma \sqrt{2T_e / \pi m_e}) + \omega_{pi}^2 [G_1(\Gamma^2 A m_p / (2k_y^2 T_i)) - k_y^2 V_{flow}^2 / \Gamma^2 G_2(\Gamma^2 A m_p / (2k_y^2 T_i))] = 0.$$

Here c represents the speed of light, k_y is the most unstable wave number perpendicular to the flow direction, V_{flow} is the flow velocity, A is the atomic number, m_p is the proton mass, Γ is the linear growth rate. Additionally, G_1 and G_2 are the dimensionless functions defined for $x > 0$ as

$$G_1(y) = 1/\sqrt{\pi} \int_0^\infty y e^{-x^2} / (x^2 + y) dx$$

$$G_2(y) = 2y/\sqrt{\pi} \int_0^\infty x^2 e^{-x^2} / (x^2 + y) dx$$

Taking experimental parameters into above dispersion equation, we obtain the relationship between the linear growth rate Γ and the wave number k_y , as shown in the Supplementary Figure 3. The most unstable mode of Weibel instability observed in our experiments is $k_y \sim 1/(L_{spacing}) = \omega_{pi}/c$, that is to say $k_y c / \omega_{pi} \sim 1$. Based on data presented in the Supplementary Fig.3(a) and 3(b), we can obtain the linear growth rate is $\Gamma \sim 0.1 V_{flow} / c \omega_{pi} \sim 1.5 \times 10^9 \text{ s}^{-1}$. All these important information about the linear growth rate of Weibel instability have been added in the supplementary information [see item 28 in

the "List of Changes"].

REVIEWERS' COMMENTS

Reviewer #1 (Remarks to the Author):

The authors have thoroughly addressed my concerns, and I now believe that this manuscript should be published in Nature Communications.

Reviewer #2 (Remarks to the Author):

The revised manuscript has adequately addressed all my previous comments and therefore should be accepted for the publication in Nature Communications.

Reviewer #3 (Remarks to the Author):

I thank the authors for preparing a thorough revision. I think the manuscript has improved and the supplemental material provided goes a long way towards addressing my original questions. I do not have additional questions.

Response Letter

Manuscript NCOMMS--23-56222

Response to Reviewer's comments

"Electron stochastic acceleration in laboratory-produced kinetic turbulent plasmas" By D.W. Yuan, Z. Lei, et.al.

We are very pleased to receive the acceptance notice. We thank again the recommendations from all the reviewers.

I. Reviewer #1:

The authors have thoroughly addressed my concerns, and I now believe that this manuscript should be published in Nature Communications.

Thanks very much for your recommendation.

II. Reviewer #2:

The revised manuscript has adequately addressed all my previous comments and therefore should be accepted for the publication in Nature Communications.

We thank the reviewer's positive comments.

III. Reviewer #3:

I thank the authors for preparing a thorough revision. I think the manuscript has improved and the supplemental material provided goes a long way towards addressing my original questions. I do not have additional questions.

We thank the reviewer for the recognition of our work.